# AUTOSIZER: Automatic Sizing of Analog and Mixed-Signal Circuits via Large Language Model (LLM) Agents

Xi Yu [1]   Dmitrii Torbunov [1]   Soumyajit Mandal [2]   Yihui Ren [1]

## Abstract

The design of Analog and Mixed-Signal (AMS) integrated circuits remains heavily reliant on expert knowledge, with transistor sizing a major bottleneck due to nonlinear behavior, high-dimensional design spaces, and strict performance constraints. Existing Electronic Design Automation (EDA) methods typically frame sizing as static black-box optimization, resulting in inefficient and less robust solutions. Although Large Language Models (LLMs) exhibit strong reasoning abilities, they are not suited for precise numerical optimization in AMS sizing. To address this gap, we propose AUTOSIZER, a reflective LLM-driven meta-optimization framework that unifies circuit understanding, adaptive search-space construction, and optimization orchestration in a closed loop. It employs a two-loop optimization framework, with an inner loop for circuit sizing and an outer loop that analyzes optimization dynamics and constraints to iteratively refine the search space from simulation feedback. We further introduce AMS-SIZINGBENCH, an open benchmark comprising 24 diverse AMS circuits in SKY130 CMOS technology, designed to evaluate adaptive optimization policies under realistic simulator-based constraints. AUTOSIZER experimentally achieves higher solution quality, faster convergence, and higher success rate across varying circuit difficulties, outperforming both traditional optimization methods and existing LLM-based agents. Our code is available at https://github.com/yuxi120407/AutoSizer.

[1]Artificial Intelligence Department, Brookhaven Laboratory, Upton, NY [2]Instrumentation Department, Brookhaven National Laboratory, Upton, NY 11973 . Correspondence to: Xi Yu <xyu1@bnl.gov>.

*Proceedings of the 43rd International Conference on Machine Learning*, Seoul, South Korea. PMLR 306, 2026. Copyright 2026 by the author(s).

## 1. Introduction

The design of Analog and Mixed-Signal (AMS) circuits typically follows a multi stage workflow that begins with topology selection and transistor sizing and then proceeds to layout level steps such as placement routing and verification while optimizing power efficiency maximizing performance and minimizing area to meet target specifications. These multi-stage workflows are complex, requiring time-consuming simulations along with human expertise. Among these stages, circuit sizing is particularly challenging because it involves a high dimensional design space and complex trade offs among multiple performance metrics, where small variations in device parameters can lead to significant and often non linear changes in circuit behavior and overall performance. As a result, automatic circuit sizing has emerged as a crucial problem in electronic design automation (EDA) and has attracted increasing research attention.

Traditional methods formulate the circuit sizing as an optimization problem, such as Bayesian Optimization (BO) (Touloupas et al., 2021; Lyu et al., 2018), Reinforcement Learning (Wang et al., 2020), and genetic algorithms (Cohen et al., 2015). However, these approaches treat the circuit design space as a static black box, ignoring domain-specific analog design expertise and often yielding suboptimal solutions. Recently, LLM-based agents have shown promise in symbolic reasoning and scientific problem solving. Several studies have explored LLMs for automated analog circuit design, focusing on circuit knowledge representation and sizing optimization. EEsizer (Liu & Chitnis, 2025b) directly uses an LLM for both circuit understanding and sizing optimization. Other LLM-agent approaches (Liu et al., 2024; Somayaji & Li, 2025) emphasize improving circuit comprehension through document parsing or structured knowledge bases. LEDRO (Kochar et al., 2025) leverages LLMs to reduce the design search space and improve optimization efficiency, while LLMACD (Xu et al., 2025) incorporates a pre-trained Transistor Behavioral Model (TBM) to enhance design consistency. Despite these advances, most existing agents rely on fixed optimization algorithms and single-stage workflows, in which the LLM interprets the circuit, applies a predetermined optimization strategy, and outputs results without iterative self-reflection or feedback-

driven refinement. This limits their ability to adapt the optimization process itself in response to intermediate outcomes, particularly for complex AMS circuits with sparse feasibility regions and competing design objectives.

To address this gap, we propose AUTOSIZER, a reflective meta-optimization framework based on multi-agent LLMs for automatic sizing optimization of analog and mixed-signal circuits. AUTOSIZER integrates circuit understanding, adaptive search-space construction, and optimization algorithm orchestration into a unified, closed-loop workflow with an evaluation agent, enabling efficient and robust exploration of high-dimensional design spaces. Starting from user specifications and circuit netlists, an LLM-based circuit understanding agent analyzes the circuit topology to identify key design variables and infer their functional roles and performance sensitivities. Based on this analysis, a search-space decision agent prioritizes the identified variables and dynamically assigns feasible ranges, enabling the framework to focus optimization effort on the most impactful parameters. Sizing optimization is carried out in a **two-loop optimization framework**. The inner loop performs **numerical circuit sizing** using a configurable optimization engine that supports multiple algorithms, such as Latin Hypercube Sampling (LHS), genetic algorithms, Bayesian Optimization (BO), and related methods, and evaluates simulation results against user-defined specifications. The outer loop performs **reflective search-space evaluation and refinement** by analyzing optimization history, convergence behavior, and constraint satisfaction. Based on this feedback, the framework adaptively revises variable priorities, tightens or expands parameter ranges, and updates circuit understanding, enabling progressive and data-efficient refinement of the design space across iterations.

To support systematic and reproducible evaluation of adaptive sizing strategies, we introduce AMS-SIZINGBENCH, a new benchmark suite for analog and mixed-signal circuit sizing optimization built entirely on the open-source SKY130 CMOS technology. The benchmark comprises of 24 representative analog and mixed-signal circuits, spanning a wide range of design complexities from basic building blocks to advanced functional modules, and is carefully organized to provide a fully open and standardized sizing environment. Each benchmark instance includes the SKY130-based circuit netlist, a clearly defined set of sizing variables and optimization constraints, and a complete simulation workflow for performance evaluation. AMS-SIZINGBENCH thus provides a fully open and standardized testbed for evaluating adaptive optimization policies and facilitates fair comparison, extensibility, and reuse in AMS sizing research.

Our contributions can be summarized as follows:

- **An LLM-driven two-loop multi-agent framework for analog and mixed-signal sizing.** We propose AUTOSIZER, a multi-agent LLM framework that that formulates AMS sizing as a meta-optimization problem, integrating circuit understanding, adaptive search-space construction, and optimization algorithm orchestration into a unified closed-loop workflow. The framework combines an inner sizing optimization loop that supports multiple optimization algorithms with an outer search-space refinement loop that analyzes optimization history, convergence behavior, and variable impact, enabling adaptive revision of variable priorities and parameter ranges for efficient exploration of high-dimensional design spaces.

- **AMS-SIZINGBENCH: an open benchmark for AMS circuit sizing systematic evaluation.** We present AMS-SIZINGBENCH, a fully open-source benchmark suite built on the SKY130 CMOS technology, comprising 24 representative analog and mixed-signal circuits spanning a wide range of design complexities. Each benchmark instance includes a standardized circuit netlist, clearly defined sizing variables and constraints, and a complete simulation and evaluation workflow, enabling reproducible evaluation and fair comparison of automated sizing methods.

- **Comprehensive experimental evaluation and performance gains.** We conduct extensive experiments on AMS-SIZINGBENCH and show that AUTOSIZER consistently outperforms both traditional optimization methods and existing LLM-based agents in terms of convergence speed, solution quality, and robustness across diverse circuit types.

**Conflict of Interest Disclosure.** The authors declare no financial conflicts of interest related to this work.

## 2. Related Work

Automated sizing of AMS circuits has been a long-standing pursuit in the EDA community, evolving from manual heuristics to simulation-driven optimization and emerging LLM-based agent workflows.

### 2.1. Traditional Circuit Sizing Methods

Traditional circuit sizing methods formulate design as a static black-box optimization problem, obtaining performance metrics through repeated simulations. Early approaches use general-purpose algorithms like simulated annealing and genetic algorithms (Wolfe & Vemuri, 2003; Pessoa et al., 2018), which require tens of thousands of simulations and lack mechanisms for adapting the search space or transferring knowledge across runs. Bayesian optimization (BO) improves sample efficiency using surrogate models (Lyu et al., 2018; Liu et al., 2014; Shahriari et al.,

*Table 1.* Comparison of LLM-based Agents circuit-sizing methods and benchmark coverage.

| Method | Self Reflection | Adaptive Optimization Tools | Search Space construction | Benchmark (Circuit Type & # Num) | | | | | | |
|---|---|---|---|---|---|---|---|---|---|---|
| | | | | Amp. | Osc. | SC | Logic/ Comp. | Ref./ Power | Filter | # Num |
| AmpAgent (Liu et al., 2024) | ✗ | ✗ | ✗ | ✓ | ✗ | ✗ | ✗ | ✗ | ✗ | 7 |
| ADO-LLM (Yin et al., 2024) | ✗ | ✗ | ✗ | ✓ | ✗ | ✗ | ✓ | ✗ | ✗ | 2 |
| LLM-USO (Somayaji & Li, 2025) | ✗ | ✗ | ✗ | ✓ | ✗ | ✗ | ✓ | ✓ | ✗ | 4 |
| LLMACD (Xu et al., 2025) | ✗ | ✗ | ✗ | ✓ | ✗ | ✗ | ✗ | ✗ | ✗ | 2 |
| LEDRO (Kochar et al., 2025) | ✓ | ✗ | ✓ | ✓ | ✗ | ✗ | ✗ | ✗ | ✗ | 22 |
| EEsizer (Liu & Chitnis, 2025b) | ✓ | ✗ | ✗ | ✓ | ✓ | ✗ | ✓ | ✗ | ✗ | 6 |
| AutoSizer | ✓ | ✓ | ✓ | ✓ | ✓ | ✓ | ✓ | ✓ | ✓ | **24** |

2015), but still relies on fixed parameter spaces and predefined acquisition strategies, and struggles as dimensionality increases. More recently, reinforcement learning (RL) methods (Wang et al., 2018; Settaluri et al., 2020; Yang et al., 2021) demonstrate improved scalability and transfer learning (Zhang et al., 2023; Ahmadzadeh et al., 2025b), yet still require hundreds of computationally expensive simulations and yield opaque, task-specific policies, limiting interpretability and hindering industrial adoption.

## 2.2. LLM Agents

Recent advances in large language models have enabled agent-based systems that move beyond passive text generation toward autonomous, goal-oriented behavior. These LLM agents leverage structured planning, tool interaction, and iterative feedback loops to decompose tasks and refine solutions (Yao et al., 2022; Chen et al., 2024). Many frameworks incorporate memory mechanisms and self-reflective reasoning to accumulate experience and adapt strategies over time. They have been successfully applied across diverse domains including software engineering (Jimenez et al., 2024), robotics (Brohan et al., 2023; Black et al., 2024), data science (Guo et al., 2024; Hong et al., 2025), and scientific research (Yu et al., 2025; Yamada et al., 2025; Baek et al., 2025). In electronic design automation, recent works explore LLM-based approaches for analog circuit design: AnalogXpert (Zhang et al., 2025) automates topology synthesis, EEschematic (Liu & Chitnis, 2025a) generates schematics via multimodal agents, AnalogCoder-Pro (Lai et al., 2026) unifies circuit generation and optimization, and SimuGen (Ren et al., 2025) constructs block diagram-based simulation models through multi-modal agents.

## 2.3. LLM-based Agents for Circuit Sizing

Recent work explores using LLMs as autonomous agents for analog circuit sizing. Unlike black-box optimization methods, LLM-based approaches leverage language models to reason about circuit structure, interpret simulation

feedback, and guide optimization.. AmpAgent (Liu et al., 2024) retrieves the amplifier-specific knowledge from essays and guides the LLM to optimize the circuit sizing. LLM-USO (Somayaji & Li, 2025) constructs a graph knowledge space and leverage it for the next circuit design. LLMACD (Xu et al., 2025) generates Transistor Behavioral Circuit Representations (TBCRs) that meet performance targets using knowledge-embedded prompts. LEDRO (Kochar et al., 2025) utilized LLMs alongside fixed optimization techniques to iteratively refine the design space for analog circuit sizing. AnaFlow (Ahmadzadeh et al., 2025a) leverages both reasoning sizing and optimizer sizing. However, existing LLM-based sizing agents rely on either fixed search spaces or fixed optimization algorithms with predetermined configurations, often validated on a single circuit class (e.g., amplifiers). This rigidity limits scalability across diverse designs. AutoSizer addresses these limitations by formulating circuit sizing as a reflective meta-optimization problem, dynamically refining search spaces and orchestrating multiple optimization algorithms based on intermediate optimization dynamics. We evaluate AutoSizer on 24 diverse AMS circuit topologies based on SKY130 CMOS technology, including OTAs, oscillators, voltage references, filters, and voltage regulators, demonstrating robust performance across varied circuit types and specifications.

## 3. Method

### 3.1. Problem Formulation

We formulate AMS circuit sizing as a simulation-driven constrained optimization problem. Let $\mathbf{x} \in \mathcal{X}$ denote the vector of sizing parameters and $\mathbf{y} = [y_1, \ldots, y_m] = f(\mathbf{x})$ denote the corresponding performance metrics obtained from simulation. A scalar figure of merit (FoM) is defined to capture the trade-off between metrics to be maximized and minimized:

$$\text{FoM}(\mathbf{y}) = \frac{\prod_{i \in \mathcal{D}} \tilde{y}_i}{\prod_{j \in \mathcal{M}} \tilde{y}_j}, \tag{1}$$

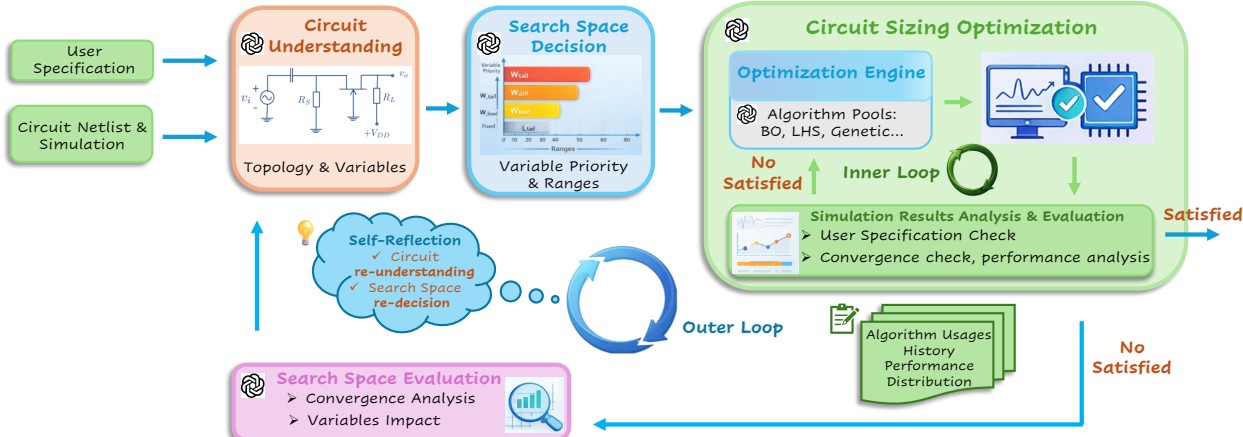

*Figure 1.* AutoSizer workflow. The framework combines LLM-based circuit understanding with an inner loop that sequentially selects, configures, and applies multiple sizing algorithms and an outer loop that adaptively refines the search space, forming a two-loop closed-loop architecture that enables iterative refinement of variable priorities and parameter ranges based on optimization feedback.

where $\mathcal{D}$ and $\mathcal{M}$ denote the sets of metrics to be maximized and minimized, respectively, and $\tilde{y}_i = y_i/y_{i,\text{spec}}$ are normalized with respect to their specification targets $y_{i,\text{spec}}$. This normalization ensures FoM values are dimensionless and comparable across circuits with different performance scales. A design is feasible when $\tilde{y}_i \geq 1$ for all $i \in \mathcal{D}$ and $\tilde{y}_j \leq 1$ for all $j \in \mathcal{M}$. The sizing task is expressed as:

$$\max_{\mathbf{x} \in \mathcal{X}} \text{FoM}(f(\mathbf{x})) \quad \text{s.t.} \quad \tilde{y}_i \geq 1 \,\forall i \in \mathcal{D}, \; \tilde{y}_j \leq 1 \,\forall j \in \mathcal{M} \tag{2}$$

### 3.2. Overview of the AutoSizer Framework

AutoSizer is a multi-agent, LLM-driven framework designed for reflective automatic sizing optimization of AMS circuits. Given user specifications and a circuit netlist, AutoSizer integrates circuit understanding, adaptive search-space construction, and simulation-driven optimization into a unified, closed-loop workflow. Multiple specialized agents collaborate to iteratively refine design decisions using simulation feedback.

As illustrated in Fig. 1, AutoSizer adopts a two-loop optimization architecture. The inner loop optimizes device sizes by invoking a configurable optimization engine and evaluating candidate designs through circuit simulation. The outer loop analyzes optimization outcomes, including convergence behavior and performance distributions, to evaluate optimization effectiveness and constraint satisfaction under the current formulation. Based on this analysis, the framework revises the optimization problem, updating variable priorities, parameter ranges, and circuit understanding through self-reflection, enabling progressive and data-efficient exploration of high-dimensional design spaces.

### 3.3. LLM-Based Circuit Understanding

The purpose of the circuit understanding module is to extract high-level structural and functional knowledge from circuit netlists to guide sizing optimization. Given a circuit netlist and user specifications as input, the LLM produces a structured summary of the circuit topology, key sizing variables, and their qualitative impact on performance metrics.

The model analyzes circuit connectivity, signal paths, and biasing structures to identify major functional blocks, and maps optimization variables to the corresponding circuit components (See circuit understanding prompt in Appendix B.1). Based on this analysis, it estimates the relative influence of sizing variables on key metrics and identifies fundamental trade-offs.

As can be seen in Figure. 2, circuit understanding agent provides an interpretable abstraction of the design space and serves as an explicit input to reflective optimization, directly informing the adaptive search-space construction stage by prioritizing impactful variables and guiding range selection.

### 3.4. Adaptive Search-Space Construction

The adaptive search-space construction module aims to reduce optimization complexity by focusing computational effort on the most impactful design parameters. Its input consists of the structured circuit understanding output and optimization dynamics and feasibility feedback from previous iterations, and its output is a dynamically defined search space with prioritized variables and feasible parameter ranges.

Directly optimizing over all possible sizing variables with wide parameter ranges leads to prohibitively large search

---

**LLM-Based Circuit Understanding Output**

**Circuit Topology:** Telescopic operational transconductance amplifier (OTA) with an NMOS differential input pair, cascoded NMOS and PMOS stages, and PMOS current-mirror loads.

**Key Sizing Variables:** $W_{\text{diff}}$ (input differential pair), $W_{\text{casc}}$ (cascode devices), $W_{\text{tail}}$ (tail current source), $W_{\text{load}}$ (PMOS load devices).

**Performance Sensitivity:** DC gain is primarily influenced by $W_{\text{diff}}$ and $W_{\text{casc}}$. UGBW is dominated by $W_{\text{diff}}$, with secondary dependence on current-scaling variables. Power consumption is mainly controlled by $W_{\text{tail}}$ and $W_{\text{load}}$.

**Key Trade-offs:** Improving gain and bandwidth increases power consumption and parasitic capacitance, requiring careful balance for FoM optimization.

---

*Figure 2.* LLM-generated circuit understanding for the telescopic OTA, including topology analysis, key sizing variables, performance sensitivities, and design trade-offs.

spaces and poor sample efficiency. In practice, only a subset of variables significantly influences circuit performance, and their effective operating ranges are often much narrower than the full technology limits. Inspired by common analog design practice, AutoSizer adopts a progressive, feedback-driven search-space expansion strategy: it begins with a compact search space defined by a small set of high-impact variables and conservative ranges (See Figure 3), and incrementally expands the search space only when optimization outcomes indicate infeasibility or stagnation.

---

**First-Round Adaptive Search-Space Construction**

**Optimization Target:** Maximize DC gain and unity-gain bandwidth (UGBW) while minimizing power consumption.

**Variable Prioritization:** $W_{\text{diff}}$ is identified as the most critical variable due to its direct control of input transconductance, which dominates both gain and UGBW. $W_{\text{tail}}$ and $W_{\text{load}}$ are ranked next, as they scale bias current and govern the primary power–performance trade-off. $W_{\text{casc}}$ is treated as a secondary knob that mainly affects gain through output resistance.

**Active Variables and Ranges:** $W_{\text{diff}} \in \{0.84, 1.26, 1.68, 2.10, 2.52\}$, $W_{\text{tail}} \in \{0.84, 1.47, 2.10, 2.52\}$, $W_{\text{load}} \in \{0.84, 1.47, 2.10, 2.52\}$. These ranges are chosen to sparsely yet effectively explore performance scaling and the FoM trade-off.

**Fixed Variable:** $W_{\text{casc}}$ is fixed at 1.89 to reduce search dimensionality while preserving high output resistance for adequate DC gain.

**Search-Space Reduction:** The original full space of $9^4 = 6561$ combinations is reduced to $5 \times 4 \times 4 = 80$ combinations, achieving an $\sim 82\times$ reduction while maintaining coverage of the dominant design trade-offs.

---

*Figure 3.* LLM-generated adaptive search space configuration for the first optimization round, showing variable prioritization, range selection, and dimensionality reduction strategy.

This reduces unnecessary exploration in early stages, focusing effort on promising regions. Upon stagnation or infeasibility, AutoSizer systematically reformulates the problem by unfixing variables or expanding ranges.

### 3.5. Two-Loop Optimization Framework

AutoSizer adopts a two-loop optimization framework that separates numerical parameter search from adaptive formulation of the optimization problem. The inner loop is responsible for optimizing sizing parameters within a fixed search space, while the outer loop evaluates the effectiveness of the current search space and revises it when necessary.

In the inner loop, AUTOSIZER adaptively selects and configures optimization algorithms from a predefined algorithm pool (See Algorithm Orchestration Prompt in Appendix B.3) based on the current optimization stage, available simulation history, and search-space characteristics determined by the LLM-based search-space decision agent. Candidate designs are evaluated via circuit simulation and scored using the specifications and FoM, with results shared across algorithms. The inner loop terminates upon convergence or budget exhaustion.

The outer loop monitors optimization outcomes produced by the inner loop, including feasibility status, convergence behavior, and performance trends. If the inner loop fails to identify feasible solutions or exhibits stagnation, the outer loop adaptively revises the optimization problem by adjusting parameter ranges, activating additional variables, or updating variable priorities. This separation of responsibilities enables efficient exploration in early stages while preserving robustness against suboptimal initial search-space assumptions. The complete two-loop optimization procedure is formally described in Algorithm 1.

---

**Algorithm 1** Two-Loop Optimization Framework

---

**Require:** Circuit netlist $\mathcal{N}$, user specifications $\mathcal{S}$, initial search space $\mathcal{X}_0$
**Ensure:** Optimized sizing parameters $\mathbf{x}^\star$
1: Initialize search space decision $\mathcal{X} \leftarrow \mathcal{X}_0$
2: Initialize optimization history $\mathcal{H} \leftarrow \emptyset$
3: **while** termination condition not met **do**
4:     **Inner Loop: Algorithm Selection & Execution**
5:     Select optimization algorithm
6:     Run optimizer $\mathcal{A}_k(\theta_k)$ within current search space $\mathcal{X}$ to generate candidate search samples
7:     **for** each candidate $\mathbf{x}$ **do**
8:         Simulate circuit and evaluate performance
9:         Update optimization history $\mathcal{H}$
10:     **end for**
11:     **Outer Loop: Search-Space Refinement**
12:     **if** feasible solution found **then**
13:         **break**
14:     **end if**
15:     **if** convergence stagnates or constraints violated **then**
16:         Analyze $\mathcal{H}$ to identify limiting variables
17:         Update search space $\mathcal{X}$ by adjusting ranges or activating new variables
18:     **end if**
19: **end while**
20: **return** Best feasible solution $\mathbf{x}^\star$

---

## 3.6. Evaluation and Feedback Mechanism

The evaluation and feedback mechanism in AUTOSIZER operates at two levels, supporting both inner-loop optimization control and outer-loop search-space refinement, as illustrated in Figure 4. Its role is to assess optimization progress, diagnose bottlenecks, and provide structured feedback for adaptive decision-making across iterations.

---

**Search Space Evaluation and Feedback**

**Optimization Status:** 5 iterations completed with 72 designs evaluated. Methods used: LHS (25 designs), BO (15 designs each in iterations 2 and 4), and Simulated Annealing (18 designs). Best FOM of 0.0990 achieved in iteration 2 and remained unchanged through iterations 3 and 4, indicating stagnation.

**Convergence Analysis:** FOM progression: [0.095, 0.099, 0.099, 0.099]. Status: converging with stagnant value at 0.099. Reason: recent improvements $< 2\%(0.00\%)$. Best FOM unchanged for 3 consecutive iterations.

**Search Space Issues (4 detected):** - $W_{\text{diff}}$: 10/10 top designs at *lower boundary* (0.84 $\mu$m) $\rightarrow$ High severity, expand lower range - $W_{\text{load}}$: 10/10 top designs at *upper boundary* (1.68 $\mu$m) $\rightarrow$ High severity, expand upper range - $W_{\text{load}}$: 10/10 top designs at *lower boundary* (1.68 $\mu$m) $\rightarrow$ High severity, expand lower range - Stagnation: Best FOM unchanged for 3 iterations (0.0990) $\rightarrow$ Medium severity

**Variable Impact Analysis:** Top design clustering: $W_{\text{casc}}$ range [1.26, 2.52] with most common values (1.68: 4$\times$, 2.1: 3$\times$, 1.26: 2$\times$). $W_{\text{diff}}$ range [0.84, 0.84] with only 0.84 appearing (10$\times$). $W_{\text{load}}$ range [1.68, 1.68] with only 1.68 appearing (10$\times$). $W_{\text{tail}}$ range [0.84, 2.52] with most common values (1.26: 3$\times$, 2.1: 3$\times$, 2.52: 2$\times$).

**Recommendations:** Priority: HIGH. Should regenerate: YES. Actions: (1) Consider stopping due to recent improvements $< 2\%(0.00\%)$, (2) Expand lower range for $W_{\text{diff}}$ based on boundary clustering and top design analysis, (3) Expand both ranges for $W_{\text{load}}$ due to dual boundary saturation, (4) Keep current ranges for $W_{\text{casc}}$ and $W_{\text{tail}}$ with adequate distribution, (5) Expand search space or change strategy to escape stagnation.

---

*Figure 4.* Search space evaluation and feedback from AutoSizer agent after inner loop finished with 4 optimization iterations.

**Inner-loop evaluation** assesses the effectiveness of the current sizing optimization strategy and guides algorithm selection and configuration. After each iteration, AutoSizer summarizes optimization outcomes using statistics such as FoM improvement, feasibility status, and parameter diversity. Based on these signals, the framework dynamically selects and configures optimization algorithms to balance exploration and exploitation, and determines termination when specifications are satisfied or convergence is detected.
**Outer-loop evaluation** assesses whether the current search space is sufficient to meet design objectives and guides high-level adaptation when necessary. It aggregates inner-loop optimization outcomes across iterations, including feasibility trends, FoM improvement trajectories, performance distributions, and parameter concentration patterns. By analyzing these signals, the outer loop identifies cases where optimization progress saturates within the current parameter bounds or where high-performing solutions consistently occur near search-space edges. In such cases, the search space is refined by adjusting parameter ranges, re-prioritizing variables, or activating previously fixed variables. By decoupling evaluation at the optimization and search-space lev-

els, AutoSizer ensures that both numerical optimization decisions and higher-level design-space refinement utilize quantitative, simulation-based evidence.

## 4. Experiments

### 4.1. Experimental Setup

**Benchmark and Evaluation.** We evaluate on AMS-SIZINGBENCH, an open SKY130-based benchmark suite consisting of 24 analog and mixed-signal circuits, including logic blocks, OTAs, oscillators, switched-capacitor circuits, voltage references, and voltage regulators, with predefined Easy, Medium, and Hard difficulty splits. Additional details of the AMS-SIZINGBENCH benchmark, including circuit descriptions, input signals, specifications, simulations, and configuration formats, are provided in Appendix A. Each circuit sizing task is a simulation-driven constrained optimization problem maximizing a figure-of-merit (FoM). Circuit performance evaluation uses Ngspice (Vogt et al., 2021) for electrical simulation with the SKY130 PDK.

**Baselines.** We compare AutoSizer against two categories of circuit sizing methods. *Traditional optimization algorithms* include Genetic Algorithm (GA) (Taherzadeh-Sani et al., 2003), Bayesian Optimization (BO), and Trust Region Bayesian Optimization (TuRBO) (Eriksson et al., 2019), representing evolutionary, surrogate-based, and trust region approaches respectively. *LLM-based agent methods* include ADO-LLM (Yin et al., 2024), LEDRO (Kochar et al., 2025), and EE-Sizer (Liu & Chitnis, 2025b). The remaining three baselines are not included due to reproducibility considerations. AmpAgent, LLM-USO, and LLM-ACD are not open-source, and the available descriptions in their respective papers lack sufficient detail for reliable reproduction.

**Implementation Details.** AutoSizer is powered by `Gemini-2.5-Flash` with a temperature of 0.4, $top\text{-}p = 0.85$, $top\text{-}k = 20$, and a maximum of 8192 output tokens. To isolate algorithmic contributions from LLM backend differences, we re-implemented LEDRO, ADO-LLM, and EE-Sizer using the same `Gemini-2.5-Flash` configuration and a unified 300-sample budget as AutoSizer. We maintain the core algorithmic logic of each method while standardizing the LLM interface and sample accounting, which enables fair comparison of optimization strategies independent of LLM-specific capabilities. Traditional baselines include: (i) a genetic algorithm (GA) with population size 20, crossover rate 0.8, mutation rate 0.1, tournament selection, and elitism; (ii) Bayesian optimization (BO) using a Gaussian Process with a Matérn kernel ($\nu = 2.5$) and UCB acquisition ($\beta = 2.0$), initialized with 10 random samples; and (iii) TuRBO with adaptive trust regions initialized at 0.8 and scaled by $2 \times / 0.5 \times$ on success/failure. All methods use a total budget of searching 300 samples. EE-Sizer runs

10 iterations with 30 LLM-generated candidates per iteration. AutoSizer uses up to three outer-loop iterations with 100 samples per inner loop, dynamically allocating samples across multiple optimization algorithms within each inner loop.

## 4.2. Overall Performance on AMS-SIZINGBENCH

Table 2 reports average three trials experimental results on the AMS-SIZINGBENCH across three difficulty levels. Performance is evaluated using Figure of Merit (FOM), number of evaluations to best solution (Evals), runtime per trial (Time), and success rate (SR), defined as the percentage of trials that meet user specifications. On the Hard benchmark, baseline LLM-based agents significantly outperform traditional optimization methods, achieving higher success rates with substantially fewer search samples. As circuit complexity increases, the design space becomes highly nonlinear and constrained; in this regime, LLM-based agents can leverage embedded circuit knowledge to guide the search more efficiently, resulting in improved robustness and efficiency. In contrast, traditional methods suffer from degraded performance and lower success rates. For the Easy and Medium benchmarks, baseline LLM-based agents exhibit comparable performance to traditional methods, indicating that when the search space is relatively simple, conventional optimization techniques remain competitive. Among the LLM-based baselines, ADO-LLM performs worse due to the absence of a self-reflection mechanism, which prevents it from guaranteeing that the LLM-generated search directions remain valid, leading to suboptimal exploration. EE-Sizer relies solely on the LLM without an explicit optimization algorithm; consequently, it requires more LLM interactions per iteration and explores a larger search space, resulting in increased runtime. LEDRO adopts a fixed optimization configuration, which limits its flexibility and reduces effectiveness, particularly for complex circuits. In contrast, our method leverages adaptive optimization strategies guided by LLM reasoning, enabling more efficient exploration of the design space and consistently superior performance across all difficulty levels, especially on complex circuit sizing tasks.

## 4.3. Ablation Studies

We systematically disable individual modules, Circuit Understanding (CU), Initial Search Space Decision (SSD), Optimization Engine (OE), and Self-Reflection Loop (SRL), and measure impact on 5 circuits (1 easy, 2 medium, 2 hard) with 3 trials each. Table 3 shows averaged results, revealing component importance and interdependencies.

**Component ranking.** Ablation results establish clear hierarchy: SRL (-39.4% FOM) > OE (-23.9%) > CU (-19.2%). This ranking demonstrates that *iterative refinement* (SRL) is

more critical than *getting initial decisions right* (CU), and *how you search* (OE) matters more than *where you search* (CU). Efficient exploration can partially compensate for suboptimal spaces, but optimal spaces cannot compensate for inefficient exploration. **SRL enables recovery from mistakes.** Removing SRL causes 80% success rate drop (100% → 20%) and 39.4% FOM loss. Single-cycle optimization is fragile because initial search space decisions may be incorrect and cannot be corrected. SRL's iterative refinement, early cycles explore broadly, later cycles exploit refined regions which is essential for robust convergence across diverse circuits. **OE improves sample efficiency.** Without OE, fixed Bayesian Optimization requires 37.5% more evaluations and 2× longer runtime, achieving only 50% success rate. Adaptive algorithm selection matches algorithms to optimization phases reducing wasted evaluations. **CU and SSD are interdependent.** A counterintuitive result: w/o CU degrades to 28.65 FOM (-19.2%), but w/o CU & SSD improves to 33.43 FOM (-5.7%). This demonstrates that **search space reduction requires accurate circuit understanding**. Without CU insights, SSD uses naming heuristics that make incorrect assumptions, constraining search to suboptimal subspaces or fixing critical variables. The full space (w/o CU & SSD) avoids these errors, outperforming misguided reduction by 16.7% despite requiring 13.9% more evaluations. **Synergistic design.** Components work synergistically to achieve best performance. Removing any component breaks this synergy, with poorly informed reduction worse than no reduction, validating the necessity of LLM-based circuit analysis.

## 4.4. Analysis and Discussion

To understand AutoSizer's optimization dynamics, we visualize search space evolution, algorithm selection and converge patterns within the inner loop, and average self-reflection circle of ring oscillator, and search space reduction across circuit difficulties, illustrated in Figure 5.

The search space evolution demonstrates strategic progressive refinement. Figure 5a shows how variable ranges adapt across three Self-Reflection Loop cycles starting from the Original full search space (blue). Loop 1 (orange) aggressively reduces the space by fixing several variables ($W_{nmos1}$, $W_{pmos2}$, $W_{nmos2}$) at specific values while keeping others active ($W_{pmos0}$, $W_{nmos0}$, $L_{inv}$). Loop 2 (green) refines further but strategically re-expands certain dimensions ($W_{pmos1}$, $W_{nmos1}$) that proved important in Loop 1, demonstrating adaptive recovery from over-aggressive reduction. Loop 3 (red) achieves the most focused search space, concentrating exploration on the critical subset of variables ($W_{nmos0}$, $W_{pmos2}$, $W_{nmos2}$) while fixing others. The non-monotonic refinement pattern where some dimensions shrink while others expand, confirming that space adaptation is data-driven based on observed performance rather than following pre-

*Table 2.* Experimental results on AMS-SIZINGBENCH. FOM, Eval, Time (s), and SR% denote *Figure of Merit*, *Evaluations to best*, *time per trial*, and *% of trials meeting user specs*, respectively. **Bold** indicates the best result and underline indicates the second best result.

| Methods | Easy | | | | Medium | | | | Hard | | | |
|---|---|---|---|---|---|---|---|---|---|---|---|---|
| | FOM↑ | Evals↓ | Time (s)↓ | SR%↑ | FOM↑ | Evals↓ | Time (s)↓ | SR%↑ | FOM↑ | Evals↓ | Time (s)↓ | SR%↑ |
| **Traditional Method** | | | | | | | | | | | | |
| Genetic | $4.0 \pm 1.2$ | $110.5 \pm 42.9$ | $515.6 \pm 28.5$ | 93.3% | $33.6 \pm 28.0$ | $193.7 \pm 47.9$ | $1638.6 \pm 18.4$ | 73.3% | $3.4 \pm 1.3$ | $165.6 \pm 49.2$ | $1689.2 \pm 65.1$ | 50.0% |
| BO | $3.2 \pm 1.5$ | $84.0 \pm 21.4$ | $415.0 \pm 55.8$ | 80.0% | $34.1 \pm 39.5$ | $153.6 \pm 59.8$ | $1492.5 \pm 48.7$ | 73.3% | $3.5 \pm 0.2$ | $151.3 \pm 51.2$ | $1875.1 \pm 73.5$ | 50.0% |
| TuRBO | $2.1 \pm 1.5$ | $20.6 \pm 6.9$ | $141.0 \pm 15.6$ | 39.8% | $30.0 \pm 47.6$ | $75.7 \pm 25.1$ | $1450.2 \pm 18.3$ | 39.8% | $2.7 \pm 1.9$ | $107.2 \pm 19.1$ | $2105.2 \pm 42.7$ | 25.0% |
| **LLM-based Agent** | | | | | | | | | | | | |
| ADO-LLM | $3.3 \pm 0.5$ | $87.2 \pm 11.5$ | $527.2 \pm 20.8$ | 80.0% | $31.9 \pm 30.5$ | $148.5 \pm 23.5$ | $1328.6 \pm 25.8$ | 80.0% | $3.5 \pm 0.7$ | $162.3 \pm 42.3$ | $1478.5 \pm 68.5$ | 41.0% |
| LEDRO | $4.1 \pm 1.1$ | $65.6 \pm 13.4$ | $385.6 \pm 15.5$ | 100% | $34.6 \pm 56.8$ | $142.5 \pm 32.5$ | $1150.7 \pm 25.4$ | 80.0% | $3.8 \pm 0.6$ | $140.8 \pm 23.2$ | $1350.8 \pm 62.5$ | 80.0% |
| EE-Sizer | $4.0 \pm 1.3$ | $95.3 \pm 21.8$ | $756.2 \pm 34.2$ | 100% | $33.3 \pm 52.9$ | $165.2 \pm 37.4$ | $1485.3 \pm 37.5$ | 80.0% | $3.7 \pm 0.9$ | $170.8 \pm 20.5$ | $1872.6 \pm 54.6$ | 56.4% |
| **AUTOSIZER** | **$4.4 \pm 1.6$** | $\underline{38.9} \pm 31.8$ | **$257.0 \pm 16.4$** | **100%** | **$36.3 \pm 58.8$** | **$42.2 \pm 51.0$** | $\underline{1359.9} \pm 29.7$ | **100%** | **$4.5 \pm 2.7$** | **$63.6 \pm 62.5$** | $\underline{1473.2} \pm 39.3$ | **100%** |

(a) Search Space Change across Outer Loop

(b) Convergence with Algorithm Selection

(c) Average Self-Reflection Cycles and Search Space Reduction Ratio

*Figure 5.* AutoSizer's optimization dynamics across three outer loops. (a) Progressive search space changes showing active variables (multiple values) and fixed variables (single value) in each outer loop of the ring oscillator. (b) Convergence trajectory with adaptive algorithm selection (LHS, BO, GA, SA) across inner loop of the ring oscillator. (c) Self-reflection loops and search space reduction ratios across circuit difficulties.

*Table 3.* Ablation study of AutoSizer on 5 representative circuits (1 easy, 2 medium, and 2 hard). CU: Circuit Understanding. SSD: Initial Search Space. OE: Optimization Engine (LLM with algorithm pool). SRL: Self-Reflection Loop.

| Methods | Components | | | | Metrics | | | |
|---|---|---|---|---|---|---|---|---|
| | CU | SSD | OE | SRL | FOM↑ | Evals↓ | Time↓ | SR%↑ |
| AutoSizer (Full) | ✓ | ✓ | ✓ | ✓ | **35.47** | **127.5** | 925.8 | **100%** |
| w/o CU | ✗ | ✓ | ✓ | ✓ | 28.65 | 137.5 | 1206.4 | 73.4% |
| w/o CU & SSD | ✗ | ✗ | ✓ | ✓ | 33.43 | 145.2 | 1053.8 | 86.8% |
| w/o OE | ✓ | ✓ | ✗ | ✓ | 27.00 | 175.4 | 1881.4 | 50.0% |
| w/o SRL | ✓ | ✓ | ✓ | ✗ | 25.32 | 150.9 | 956.8 | 50.0% |
| w/o CU & SSD & SRL | ✗ | ✗ | ✓ | ✗ | 21.51 | 136.5 | 857.6 | 20.0% |

determined monotonic reduction, enabling the system to recover from suboptimal early decisions.

The convergence curves show consistent improvement across refinement cycles. Figure 5b demonstrates that each outer loop starts with higher FOM than the previous cycle's starting point and achieves progressively better final performance. Within each cycle, the system saves optimization history at every iteration, enabling the LLM to make informed decisions about algorithm selection and configuration based on observed convergence patterns. This pattern

where later cycles both start better and converge faster validates that Self-Reflection Loop's search space decisions are correct: refined spaces eliminate unpromising regions, enabling each cycle to build upon previous insights rather than restarting from scratch.

Circuit difficulty affects iteration requirements but not reduction effectiveness. Figure 5c shows that harder circuits require more self-reflection outer loop due to increased specification complexity and sparser feasible design regions. These additional iterations are needed to progressively refine the search space through variable unfixing and range expansion when initial configurations prove insufficient. However, search space reduction remains consistent across all difficulty levels, demonstrating that LLM-based circuit analysis effectively identifies high-impact variables regardless of topology complexity.

### 4.5. LLM Strategy Decision Analysis

To validate that AutoSizer's LLM-based strategy selection produces non-trivial, adaptive decisions rather than following a fixed pattern, we logged every strategy decision across optimization cycles. Table 4 shows the complete decision

trace for the bandgap reference circuit. Three adaptive behaviors emerge that no fixed or random schedule can replicate, summarized in Table 5.

*Table 4.* LLM strategy decision trace for the bandgap circuit across three optimization cycles.

|  | Cycle 0 (3 vars) | Cycle 1 (5 vars) | Cycle 2 (4 vars) |
|---|---|---|---|
| Iter 0 | LHS $n$=15 | BO $n$=35 | BO $n$=25 |
| Iter 1 | Genetic $n$=15 | BO $n$=50 | BO $n$=35 |
| Iter 2 | BO $n$=15 | LHS $n$=50 (re-explore) | — |
| Iter 3 | SA $n$=15 | — | — |
| Iter 4 | LHS $n$=50 (re-explore) | — | — |
| #Evals | 110 | 135 | 60 |
| Vars | 3 | 5 | 4 |
| Best FOM | 0.1085 | 0.1152 (+6.2%) | **0.1209** (+11.5%) |

*Table 5.* Adaptive behaviors observed in LLM-guided strategy selection and why fixed/random schedules cannot replicate them.

| Adaptive behavior | What the LLM did | Why fixed/random fails |
|---|---|---|
| Mid-cycle re-exploration | Detected stagnation in Cycle 1, switched back to LHS | Fixed never re-explores; random re-explores arbitrarily |
| Circuit-specific selection | Bandgap: 4 methods incl. SA. Switched-cap: skipped SA, alternated BO↔LHS | One fixed schedule cannot fit different FOM landscapes |
| Dimension-aware sizing | $n$=15 for 3 vars → $n$=35–50 for 5 vars → $n$=25–35 for 4 vars | Fixed uses constant $n$ regardless of space size |

## 4.6. Search-Space Adaptation Analysis

To analyze what drives search-space transitions, we examine the five categories of feedback signals that the LLM receives at each regeneration cycle boundary (Table 6). These signals are programmatically extracted from the inner optimization loop with no manual curation.

*Table 6.* Feedback signals provided to the LLM at each outer-loop regeneration cycle boundary.

| Signal | What the LLM sees | Example trigger |
|---|---|---|
| Performance progression | FOM history across all iterations | Stagnation triggers expand or unfix |
| Convergence status | Whether FOM improved in last 2 iterations, with stagnation flag | `stagnant=True` → variable unfixing |
| Variable impact | Which variables show variance vs. convergence in top designs | 70%+ converge to one value → fix it, unfix another |
| Boundary detection | Whether top designs cluster at range edges | 20%+ at boundaries → expand ranges |
| Top design details | Full parameter values and metrics of best $N$ designs | Reveals true performance drivers |

Based on these signals, the LLM selects one of six actions: *continue current*, *expand ranges*, *narrow ranges*, *unfix variables*, *change focus*, or *converged*. Table 7 reports the frequency of each action, its average FOM impact, and the number of iterations before the action is triggered, stratified by circuit difficulty.

The *expand ranges* or *unfix variables* action grows from

*Table 7.* Outer-loop action frequency, FOM impact, and timing across circuit difficulty levels.

| Diff. | Action | Freq. | Avg FOM △ | Iters before (mean±std) |
|---|---|---|---|---|
| Easy | expand / unfix | 36.4% | +1.5% | 104±41 |
|  | narrow_ranges | 6.0% | +0.5% | 164±32 |
|  | change_focus | 7.1% | +0.3% | 154±63 |
|  | continue_current | 50.5% | +0.0% | 125±64 |
| Med. | expand / unfix | 52.4% | +2.5% | 104±80 |
|  | narrow_ranges | 20.7% | +1.3% | 118±21 |
|  | change_focus | 16.3% | +0.3% | 155±36 |
|  | continue_current | 10.6% | +0.1% | 191±80 |
| Hard | expand / unfix | 57.1% | +16.4% | 105±40 |
|  | narrow_ranges | 10.0% | −3.8% | 180±48 |
|  | change_focus | 20.0% | +2.9% | 183±52 |
|  | continue_current | 12.9% | +1.6% | 197±35 |

36.4% on easy circuits to 57.1% on hard ones, with FOM impact rising from +1.5% to +16.4%, while *continue current* dominates easy circuits (50.5%), confirming the inner loop alone suffices for low-dimensional problems. On harder circuits, the LLM increasingly reshapes the search space. The "Iters before" column reveals a principled ordering: expansion occurs early, while narrowing and refocusing occur later, reflecting an exploration-first strategy that emerges from optimization feedback rather than hard-coded rules.

## 5. Conclusion

In this work, we propose AUTOSIZER, a reflective meta-optimization framework based on LLMs-based agents for automatic analog and mixed-signal circuit sizing. Additionally, we introduce AMS-SIZINGBENCH, a comprehensive open-source benchmark for evaluating LLM-based agents on analog and mixed-signal circuit sizing that includes SKY130-based circuit netlists, a well-defined set of sizing variables, and a complete simulation workflow for performance evaluation. Comprehensive experiments on the proposed AMS-SIZINGBENCH demonstrate that AUTOSIZER consistently outperforms prior methods in both success rate and figure of merit, particularly on hard, constraint-tight designs. We hope that AUTOSIZER and AMS-SIZINGBENCH provide a foundation for future research in reliable, scalable LLM-driven AMS circuit design automation.

## Acknowledgments

This work was supported by the U.S. Department of Energy (DOE) Office of Science, Office of Advanced Scientific Computing Research (ASCR), through an ASCR EXPRESS award (DE-SCL0000083), and by the Laboratory Directed Research and Development (LDRD) Program (LDRD 25-041 and LDRD 25-006) at Brookhaven National Laboratory, which is operated and managed for the DOE Office of Science by Brookhaven Science Associates under Contract No.

DE-SC0012704. We also express our sincere gratitude to Dr. Yuewei Lin and Dr. Shinjae Yoo at Brookhaven National Laboratory for their valuable advice and feedback.

## Impact Statement

This paper presents work whose goal is to advance the fields of Electronic Design Automation and Machine Learning through automated analog circuit design. We note that our framework is designed to augment rather than replace human circuit designers, with interpretable LLM-generated reasoning serving as an educational and verification tool. Users should validate AutoSizer-generated designs through standard verification flows before fabrication. The release of AMS-SizingBench as an open-source benchmark promotes reproducibility and accessibility in circuit optimization research, supporting both academic study and practical deployment of AI-assisted design automation.

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

# A. Details of AMS-SIZINGBENCH

## A.1. AMS-SIZINGBENCH Benchmark Circuit Summary

*Table 8.* AMS-SIZINGBENCH Benchmark Circuit Summary

| Circuit | Type | Diff. | #T | Description | Circuit | Type | Diff. | #T | Description |
|---|---|---|---|---|---|---|---|---|---|
| inverter | Logic | Easy | 2 | Basic CMOS logic gate that inverts the input signal. | five_trans_ota | Amp | Med. | 6 | Compact operational transconductance amplifier with minimal transistor count. |
| buffer | Logic | Easy | 4 | Cascaded inverter structure for restoring logic levels. | vco | Osc. | Med. | 10 | Voltage-controlled oscillator with frequency tunable by control voltage. |
| nand_gate | Logic | Easy | 4 | CMOS logic gate implementing the NAND Boolean function. | telescopic_ota | Amp | Med. | 10 | High-gain telescopic cascode OTA using stacked devices. |
| resistive_load_amp | Amp | Easy | 1 | Single-transistor amplifier using a resistive load. | current_mirror_ota | Amp | Med. | 10 | OTA based on current-mirror biasing for gain and matching. |
| diode_load_amp | Amp | Easy | 2 | Amplifier employing a diode-connected load. | folded_cascode_ota | Amp | Med. | 10 | Folded cascode OTA enabling wide input common-mode range. |
| ring_oscillator (3-stage) | Osc. | Med. | 6 | Three-stage inverter loop generating oscillation. | ldo | Power | Hard | 11 | Low-dropout voltage regulator operating under limited headroom. |
| bandgap | Ref. | Hard | 12 | Temperature-stable voltage reference circuit. | folded_cascode_ota_low_pass_filter | Filter | Hard | 10 | Low-pass active filter implemented using a Sallen–Key topology with a folded-cascode OTA. |
| switched_capacitor | SC | Hard | 10 | Switched-capacitor circuit with sampling network and OTA. | folded_cascode_ota_high_pass_filter | Filter | Hard | 10 | High-pass active filter implemented using a Sallen–Key topology with a folded-cascode OTA. |
| xor_gate | Logic | Hard | 14 | CMOS logic gate implementing the XOR operation. | folded_cascode_ota_band_pass_filter | Filter | Hard | 10 | Band-pass OTA-based active filter implemented using a Sallen–Key topology with a folded-cascode OTA. |
| AZC | Amp | Hard | 25 | Active-Zero Compensation technique for offset and low-frequency error reduction. | fdgb | Amp | Hard | 26 | Fully Differential Gain-Boosting Amplifier architecture. |
| nmcnr | Amp | Hard | 24 | Nested Miller Compensation with Nulling Resistor for stability enhancement. | smc | Amp | Hard | 24 | Simple Miller Compensation three-stage amplifier. |
| dfcfc | Amp | Hard | 26 | Damping Factor Control Frequency Compensation for multi-stage amplifiers. | iac | Amp | Hard | 35 | Indirect Amplifier Compensation technique for high-speed stability. |

## A.2. AMS-SIZINGBENCH Example

```
pdk_lib_path: ".../skywater-pdk-libs-sky130_fd_pr/combined_models/sky130.lib.spice"
align_pdk_path: ".../ALIGN-pdk-sky130/SKY130_PDK/"
pex_script_path: ".../ALIGN-public/test_magic_pex.py"
results_dir: "./telescopic_ota_test_folder"

user_specs: "Primary goal: Maximize DC gain and UGBW while minimizing power consumption
    for a telescopic OTA with 1pF load."
user_specs_metric: "fom > 0.100 AND dc_gain_db > 55 AND ugbw > 10 AND power_dc < 50"

params:
  L: 0.15
  vdd: 1.8
  vcm: 0.7
  cload: 1e-12
  ibias: 10e-6
  vbiasp1: 1.15

variable:
  W_tail_base: null
  W_diff_base: null
  W_casc_base: null
  W_load_base: null

W_values: [0.84, 1.05, 1.26, 1.47, 1.68, 1.89, 2.10, 2.31, 2.52]
```

```yaml
width_scales:
  W_tail: [W_tail_base, 2]
  W_diff: [W_diff_base, 4]
  W_casc: [W_casc_base, 2]
  W_load: [W_load_base, 2]

subckt_name: TELESCOPIC_OTA
subckt_pins:
  - VBIASN
  - VBIASNC
  - VBIASP1
  - VBIASP2
  - VINN
  - VINP
  - VOUTN
  - VOUTP
  - VDD
  - "0"

testbench_signals:
    VBIASN: VBIASN
    VBIASNC: VBIASNC
    VBIASP1: VBIASP1
    VBIASP2: VBIASP2
    VINN: VINN
    VINP: VINP
    VOUTN: VOUTN
    VOUTP: VOUTP
    VDD: VDD
    "0": "0"

metrics:
  - dc_gain_db
  - ugbw
  - power_dc
  - fom

base_metrics:
  gain_db:
    name: "Gain"
    unit: "dB"
    format: ".1f"
    degradation_key: "gain_percent"

  power_uw:
    name: "Power"
    unit: "µW"
    format: ".1f"
    degradation_key: "power_percent"

  ugbw_mhz:
    name: "UGBW"
    unit: "MHz"
    format: ".1f"
    degradation_key: "ugbw_percent"

  fom:
    name: "FOM"
    unit: ""
    format: ".3f"
    degradation_key: "fom_percent"

ota_subckt_template: |
    .subckt TELESCOPIC_OTA VBIASN VBIASNC VBIASP1 VBIASP2 VINN VINP VOUTN VOUTP VDD 0
    * Tail current sources
```

```
    xm1 ID VBIASN 0 0 sky130_fd_pr__nfet_01v8 w={W_tail} l={L}
    xm2 NET10 VBIASN 0 0 sky130_fd_pr__nfet_01v8 w={W_tail} l={L}
    * Differential pair
    xm3 NET8 VINP NET10 0 sky130_fd_pr__nfet_01v8 w={W_diff} l={L}
    xm4 NET014 VINN NET10 0 sky130_fd_pr__nfet_01v8 w={W_diff} l={L}
    * NMOS cascode (use VBIASNC here)
    xm5 VOUTN VBIASNC NET8 0 sky130_fd_pr__nfet_01v8 w={W_casc} l={L}
    xm6 VOUTP VBIASNC NET014 0 sky130_fd_pr__nfet_01v8 w={W_casc} l={L}
    * PMOS cascode
    xm7 VOUTN VBIASP1 NET06 VDD sky130_fd_pr__pfet_01v8 w={W_casc} l={L}
    xm8 VOUTP VBIASP1 NET012 VDD sky130_fd_pr__pfet_01v8 w={W_casc} l={L}
    * PMOS current source
    xm9 NET06 VBIASP2 VDD VDD sky130_fd_pr__pfet_01v8 w={W_load} l={L}
    xm10 NET012 VBIASP2 VDD VDD sky130_fd_pr__pfet_01v8 w={W_load} l={L}
    .ends TELESCOPIC_OTA

testbench_template: |
  * Telescopic OTA Simulation
    .lib {pdk_lib_path} tt
    {ota_subckt}
    * Power supply
    VDD VDD 0 DC {vdd}
    * Bias Generation
    IBIAS_N VDD VBIASN DC {ibias}
    XMN_BIAS VBIASN VBIASN 0 0 sky130_fd_pr__nfet_01v8 w={W_tail} l={L}
    IBIAS_P VBIASP2 0 DC {ibias}
    XMP_BIAS VBIASP2 VBIASP2 VDD VDD sky130_fd_pr__pfet_01v8 w={W_load} l={L}
    VBIASP1 VBIASP1 0 DC={vbiasp1}
    IBIAS_NC VDD VBIASNC DC {ibias}
    XMN_BIASC VBIASNC VBIASNC 0 0 sky130_fd_pr__nfet_01v8 w={W_casc} l={L}
    * Differential inputs
    VINP VINP 0 DC {vcm} AC 0.5
    VINN VINN 0 DC {vcm} AC -0.5
    * OTA instance
    XOTA {inst_pins} {subckt_name}
    * Load capacitors
    CLOADP VOUTP 0 {cload}
    CLOADN VOUTN 0 {cload}

    .op
    .control
    op
    * Power
    let vdd_current = abs(i(VDD))
    let power_dc = vdd_current * {vdd}
    echo "=== POWER ==="
    print vdd_current power_dc
    * AC analysis
    ac dec 100 1 10g

    let vout_diff = v(VOUTP) - v(VOUTN)
    let gain_db = db(vout_diff)
    let dc_gain_db = gain_db[0]

    echo "=== DC GAIN ==="
    print dc_gain_db

    * UGBW
    if dc_gain_db > 0
       meas ac ugbw WHEN gain_db=0 CROSS=1
    end
    quit
    .endc
    .end
```

## A.3. Sizing Variables and Specification

This section provides the details of each circuit in the AMS-SIZINGBENCH input and output specifications, and the simulation checklist used to validate each circuit block against design requirements.

*Table 9.* Circuit blocks: inputs, output specifications, and simulation checklist.

| Circuit name | Input signal(s) | Output specification(s) | Simulation part (what to run) |
|---|---|---|---|
| Inverter | $V_{DD} = 1.8\,\text{V}$; $V_{IN}$ DC sweep from 0 to 1.8 V with 10 mV step and rail-to-rail pulse with 100 ps rise/fall time; load capacitance $C_L = 10\,\text{fF}$ | Maximum DC voltage gain (dB); average propagation delay at 50% threshold ($t_{PLH}$, $t_{PHL}$); dynamic power consumption. | DC sweep for transfer characteristic and gain extraction; transient analysis for delay and power; static and dynamic power separation using supply current; transistor width sweep for performance optimization |
| Buffer | $V_{DD} = 1.8\,\text{V}$; $V_{IN}$ rail-to-rail pulse from 0 to 1.8 V with 100 ps rise/fall time and 4 ns period; DC sweep from 0 to 1.8 V with 10 mV step; load capacitance $C_L = 10\,\text{fF}$ | Maximum DC voltage gain (dB); average propagation delay at 50% threshold ($t_{PLH}$, $t_{PHL}$); dynamic power consumption. | DC sweep for overall buffer gain extraction; transient analysis for propagation delay and switching behavior; average and static power estimation from supply current; dynamic power computed as average minus static power; transistor width sweep across both stages for performance optimization |
| NAND (2-input) | $V_{DD} = 1.8\,\text{V}$; two digital inputs $A$ and $B$ driven by piecewise-linear waveforms to exercise NAND switching cases, including $B$ rising and falling while $A$ is held high; rise/fall time 100 ps; operating frequency 100 MHz; load capacitance $C_L = 10\,\text{fF}$ | Propagation delays $t_{PHL}$ and $t_{PLH}$ measured at 50% $V_{DD}$ threshold; average propagation delay; static, dynamic, and total power consumption; energy per transition; output voltage levels $V_{OH}$ and $V_{OL}$; noise margins (NMH, NML) | Operating-point analysis for static current and static power; transient simulation for delay, dynamic power, and energy-per-transition extraction; delay measured from input $B$ to output $Y$ at 50% $V_{DD}$ crossings; output voltage extrema used to compute $V_{OH}$, $V_{OL}$, and noise margins; |
| Resistive-Load Amplifier (Common-Source + $R_L$) | $V_{DD} = 1.8\,\text{V}$; input $V_{IN}$ biased at $V_{\text{bias}} = 0.6\,\text{V}$ with small-signal excitation $v_{ac} = 1\,\text{mV}$; load $R_L = 50\,\text{k}\Omega$ and $C_L = 10\,\text{fF}$; AC sweep from 1 Hz to 10 GHz | Small-signal voltage gain (dB); $-3\,\text{dB}$ bandwidth (MHz); DC power consumption ($\mu$W); and output voltage swing (V) | Operating-point analysis to extract $I_{DD}$ and $V_{OUT,DC}$; AC analysis (decade sweep) to obtain gain response and determine $-3\,\text{dB}$ bandwidth; power computed from $I_{DD}V_{DD}$; output swing estimated from the DC bias point. |
| Diode-Load Amplifier (Common-Source + Diode-Connected Load) | $V_{DD} = 1.8\,\text{V}$; input $V_{IN}$ biased at 0.6 V with small-signal AC excitation of 1 mV; load capacitance $C_L = 10\,\text{fF}$; AC frequency sweep from 1 Hz to 10 GHz | Small-signal voltage gain (dB); $-3\,\text{dB}$ bandwidth (MHz); DC power consumption ($\mu$W); output voltage swing (V) | Operating-point analysis to extract supply current and DC output voltage; AC analysis to obtain gain response and determine $-3\,\text{dB}$ bandwidth; power computed from $I_{DD}V_{DD}$; output swing estimated from the DC operating point; parametric sweep of input and load transistor dimensions |
| 3-stage ring oscillator | $V_{DD} = 1.8\,\text{V}$; three cascaded CMOS inverters connected in a ring; initial condition applied to one node for startup; load capacitance $C_L = 10\,\text{fF}$; transient simulation | Oscillation frequency (MHz); oscillation period (ns); average power consumption ($\mu$W); delay per inverter stage (ps) | Transient simulation to capture steady-state oscillation; period measured from consecutive rising zero-crossings at 50% $V_{DD}$; oscillation frequency computed from period; average power calculated from mean supply current; delay per stage extracted as period/$(2N)$ for $N = 3$ stages |
| Bandgap reference | Supply voltage $V_{DD}$ with nominal value 2.0 V; DC supply sweep from 1.8 V to 3.3 V for line regulation measurement; AC small-signal excitation at the supply for PSRR analysis; temperature set to 27°C | Reference voltage $V_{\text{REF}}$ (V); DC power consumption ($\mu$W); line regulation (%/V); power supply rejection ratio (PSRR, dB) | Operating-point analysis to extract $V_{\text{REF}}$ and supply current; DC sweep of $V_{DD}$ to evaluate line regulation; AC analysis with supply perturbation to extract PSRR; power computed from supply current and nominal $V_{DD}$ |
| Current-Mirror OTA | $V_{DD} = 1.8\,\text{V}$; tail bias current $I_{\text{BIAS}} = 20\,\mu\text{A}$; differential input with common-mode voltage $V_{\text{CM}} = 0.9\,\text{V}$ and small-signal excitation ($+0.5 / -0.5$ AC); load capacitance $C_L = 1\,\text{pF}$ | DC voltage gain (dB); unity-gain bandwidth (UGBW, MHz); DC power consumption ($\mu$W) | Operating-point analysis to extract supply current and DC power; AC analysis to obtain differential gain response; DC gain extracted at low frequency; unity-gain bandwidth measured at 0 dB gain crossing; |
| Five-transistor OTA | $V_{DD} = 1.8\,\text{V}$; tail bias current $I_{\text{BIAS}} = 20\,\mu\text{A}$; differential input with common-mode voltage $V_{\text{CM}} = 0.9\,\text{V}$ and small-signal excitation ($+0.5 / -0.5$ AC); load capacitance $C_L = 1\,\text{pF}$ | DC voltage gain (dB); unity-gain bandwidth (UGBW, MHz); DC power consumption ($\mu$W). | Operating-point analysis to extract supply current and DC power; AC analysis to obtain differential gain response; DC gain extracted at low frequency; unity-gain bandwidth measured at 0 dB gain crossing; |

**AutoSizer: Automatic Sizing of Analog and Mixed-Signal Circuits via Large Language Model (LLM) Agents**

| Circuit name | Input signal(s) | Output specification(s) | Simulation part (what to run) |
|---|---|---|---|
| Telescopic cascode OTA | $V_{DD}$ = 1.8 V; tail bias current $I_{\text{BIAS}}$ = 10 μA; differential input with common-mode voltage $V_{\text{CM}}$ = 0.7 V and small-signal excitation (+0.5 / −0.5 AC); bias voltages $V_{\text{BIASN}}$, $V_{\text{BIASNC}}$, $V_{\text{BIASP1}}$ = 1.15 V, and $V_{\text{BIASP2}}$; differential load capacitance $C_L$ = 1 pF per output | DC differential voltage gain (dB); unity-gain bandwidth (UGBW, MHz); DC power consumption (μW) | Operating-point analysis to establish bias currents and extract DC power; AC analysis of differential output to obtain gain response; DC gain extracted at low frequency; unity-gain bandwidth measured at 0 dB differential gain crossing; parametric sweep of tail, input, cascode, and load device widths to evaluate gain–bandwidth–power trade-offs |
| Switched-Capacitor Circuit | $V_{DD}$ = 1.8 V; differential OTA biased with $I_{\text{BIAS}}$ = 10 μA; input signal $V_{IN}$ with DC bias around mid-supply and AC amplitude 0.25 V; two non-overlapping clock phases at 1 MHz; sampling, holding, and load capacitors ($C_{\text{samp}}$, $C_{\text{hold}}$, $C_{\text{load}}$) | Closed-loop gain (dB) and gain error; settling time (ns); charge injection (mV); DC power consumption (μW); total harmonic distortion (THD, dB); output voltage swing (V); unity-gain bandwidth (UGBW, MHz); DC gain (dB); phase margin (deg) | Transient simulation of a switched-capacitor integrator using two non-overlapping clock phases at 1 MHz; operating-point analysis to obtain DC bias and initial output swing; closed-loop gain extracted from steady-state output voltage after multiple clock cycles; settling time measured using a 1% error criterion relative to final value; charge injection quantified from voltage disturbance at the sampling node during clock transitions; average supply current used to compute DC power consumption; FFT-based spectral analysis of the output waveform to extract total harmonic distortion (THD); separate open-loop OTA AC analysis performed to extract DC gain, unity-gain bandwidth (UGBW), and phase margin |
| Low-Dropout Regulator (LDO) | Supply voltage $V_{DD}$ = 1.8 V with DC sweep from $0.9V_{DD}$ to $1.1V_{DD}$; reference voltage $V_{\text{REF}}$ = 0.6 V; programmable load current swept from $0.1I_{\text{LOAD}}$ to $2I_{\text{LOAD}}$ (up to 20 mA); bias current $I_{\text{BIAS}}$ = 10 μA | Regulated output voltage $V_{\text{OUT}}$ (V); dropout voltage (mV); line regulation (mV/V); load regulation (mV/mA); power supply rejection ratio (PSRR, dB); DC power consumption (μW); output voltage variation over temperature (V) | Operating-point analysis to establish nominal bias and output voltage; DC supply-voltage sweep from $0.9V_{DD}$ to $1.1V_{DD}$ to extract dropout voltage and line regulation; DC load-current sweep from $0.1I_{\text{LOAD}}$ to $2I_{\text{LOAD}}$ to evaluate load regulation; AC analysis with small-signal supply injection to extract PSRR; DC power consumption computed from measured supply and load currents |
| Voltage-Controlled Oscillator | Supply voltage $V_{DD}$ = 1.8 V; control voltage $V_{\text{CTRL}}$ swept from 0 to 1.8 V; five-stage CMOS ring oscillator with inverter-based delay cells; temperature set to 27°C | Oscillation frequency (Hz); DC power consumption (μW); tuning range (%); frequency at minimum and maximum control voltage; VCO gain (MHz/V) | Transient simulation of a five-stage CMOS ring oscillator with voltage-dependent load capacitance to induce frequency tuning; a single control-voltage run used to extract steady-state oscillation frequency and DC power consumption from time-domain waveforms; oscillation frequency measured from rising zero-crossings after steady-state is reached; power computed from the median supply current during steady-state operation; extended characterization performed by sweeping the control voltage across the specified range and repeating transient simulations at each point to extract frequency tuning behavior; tuning range computed from minimum and maximum oscillation frequencies; average VCO gain (MHz/V) extracted from local frequency–voltage slopes; statistical filtering applied to remove high-power outliers before aggregating power metrics; overall performance summarized using a composite figure of merit (FOM) |
| XOR Gate (2-Input CMOS XOR with Transmission Gates) | $V_{DD}$ = 1.8 V; two digital inputs $A$ and $B$ driven by piecewise-linear waveforms to exercise XOR switching conditions, including rising and falling transitions of $B$ while $A$ is held high; rise/fall time 100 ps; operating frequency 100 MHz; load capacitance $C_L$ = 10 fF | Propagation delays $t_{PHL}$ and $t_{PLH}$ (ps); average propagation delay (ps); static, dynamic, and total power consumption (μW); energy per transition (fJ); output voltage levels $V_{OH}$ and $V_{OL}$ (V); noise margins (NMH, NML) | Operating-point analysis to extract static supply current and static power; transient simulation with representative input transitions to measure propagation delays at the 50% $V_{DD}$ threshold; average dynamic power computed from mean supply current during switching activity; energy per transition calculated from dynamic power and switching frequency; output voltage extrema used to extract $V_{OH}$, $V_{OL}$, and noise margins |
| Active Zero Compensated OTA (AZC) | $V_{DD}$ = 1.8 V; differential input with $V_{\text{CM}}$ = 0.3 V; small-signal AC excitation (+0.5/ −0.5); bias current $I_{\text{BIAS}}$ = 1 μA; load $R_L$ = 25 kΩ, $C_L$ = 15 nF | DC voltage gain (dB); unity-gain bandwidth (UGBW); DC power consumption (μW). | Operating-point analysis to extract supply current and DC power; AC analysis (1 Hz–10 GHz) with differential excitation to obtain gain response; DC gain extracted from low-frequency AC magnitude; unity-gain bandwidth measured at 0 dB gain crossing; |
| Damping-Factor-Controlled Folded-Cascode OTA (DFCFC) | $V_{DD}$ = 1.8 V; differential input with $V_{\text{CM}}$ = 0.9 V; small-signal AC excitation (+0.5/ −0.5); bias current $I_{\text{BIAS}}$ = 10 μA; load $R_L$ = 25 kΩ, $C_L$ = 15 nF | DC voltage gain (dB); unity-gain bandwidth (UGBW); DC power consumption (μW). | Operating-point analysis to establish bias currents and compute DC power from supply current; AC analysis (1 Hz–10 GHz) with differential excitation to obtain gain response; DC gain extracted from low-frequency AC magnitude; unity-gain bandwidth measured at 0 dB gain crossing; |

**AutoSizer: Automatic Sizing of Analog and Mixed-Signal Circuits via Large Language Model (LLM) Agents**

| Circuit name | Input signal(s) | Output specification(s) | Simulation part (what to run) |
|---|---|---|---|
| Fully Differential Gain-Boosted OTA (FDGB) | $V_{DD} = 1.8\,\text{V}$; fully differential input with $V_{\text{CM}} = 0.9\,\text{V}$; small-signal AC excitation ($+0.5/-0.5$); bias current $I_{\text{BIAS}} = 10\,\mu\text{A}$; differential load $R_L = 25\,\text{k}\Omega$, $C_L = 15\,\text{nF}$ per output | DC differential voltage gain (dB); unity-gain bandwidth (UGBW); DC power consumption ($\mu\text{W}$). | Operating-point analysis to establish bias currents and compute DC power from supply current; AC analysis (1 Hz–10 GHz) with fully differential excitation and differential output sensing ($V_{OUTP} - V_{OUTN}$); DC gain extracted from low-frequency differential AC magnitude; unity-gain bandwidth measured at 0 dB differential gain crossing. |
| Indirectly Compensated OTA (IAC) | $V_{DD} = 1.8\,\text{V}$; differential input with $V_{\text{CM}} = 0.9\,\text{V}$; small-signal AC excitation ($+0.5/-0.5$); bias current $I_{\text{BIAS}} = 47\,\mu\text{A}$; load $R_L = 25\,\text{k}\Omega$, $C_L = 15\,\text{nF}$ | DC voltage gain (dB); unity-gain bandwidth (UGBW); DC power consumption ($\mu\text{W}$). | Operating-point analysis to establish bias currents and compute DC power from supply current; AC analysis (1 Hz–10 GHz) with differential excitation to obtain gain response; DC gain extracted from low-frequency AC magnitude; unity-gain bandwidth measured at 0 dB gain crossing; compensation behavior governed by indirect Miller network. |
| Nested-Miller OTA with Nulling Resistor (NM-CNR) | $V_{DD} = 1.8\,\text{V}$; differential input with $V_{\text{CM}} = 0.9\,\text{V}$; small-signal AC excitation ($+0.5/-0.5$); bias current $I_{\text{BIAS}} = 19\,\mu\text{A}$; load $R_L = 25\,\text{k}\Omega$, $C_L = 15\,\text{nF}$ | DC voltage gain (dB); unity-gain bandwidth (UGBW); DC power consumption ($\mu\text{W}$). | Operating-point analysis to establish bias currents and compute DC power from supply current; AC analysis (1 Hz–10 GHz) with differential excitation to obtain gain response; DC gain extracted from low-frequency AC magnitude; unity-gain bandwidth measured at 0 dB gain crossing; nested Miller compensation with nulling resistor used for pole–zero placement. |
| Single-Miller-Compensated OTA (SMC) | $V_{DD} = 1.8\,\text{V}$; differential input with $V_{\text{CM}} = 0.9\,\text{V}$; small-signal AC excitation ($+0.5/-0.5$); bias current $I_{\text{BIAS}} = 20\,\mu\text{A}$; load $R_L = 25\,\text{k}\Omega$, $C_L = 15\,\text{nF}$ | DC voltage gain (dB); unity-gain bandwidth (UGBW); DC power consumption ($\mu\text{W}$). | Operating-point analysis to establish bias currents and compute DC power from supply current; AC analysis (1 Hz–10 GHz) with differential excitation to obtain gain response; DC gain extracted from low-frequency AC magnitude; unity-gain bandwidth measured at 0 dB gain crossing; dominant-pole stabilization achieved via single Miller compensation capacitor; FOM computed from extracted gain, UGBW, and power. |
| Folded-cascode OTA | $V_{DD} = 1.8\,\text{V}$; tail current $I_{\text{TAIL}} = 10\,\mu\text{A}$; differential input with common-mode voltage $V_{\text{CM}} = 0.9\,\text{V}$ and small-signal excitation ($+0.5 / -0.5$ AC); bias voltages $V_{BN} = 0.7\,\text{V}$ and $V_{BP} = 1.0\,\text{V}$; load capacitance $C_L = 1\,\text{pF}$ | DC voltage gain (dB); unity-gain bandwidth (UGBW, MHz); DC power consumption ($\mu\text{W}$). | Operating-point analysis to establish bias currents and extract DC power; AC analysis to obtain differential gain response; DC gain measured as maximum low-frequency gain; unity-gain bandwidth extracted at 0 dB gain crossing; parametric sweep of device widths to optimize gain–bandwidth–power trade-offs under a fixed 1 pF load. |
| Folded-Cascode OTA with Integrated Low-Pass Filter (LPF) | $V_{DD} = 1.8\,\text{V}$; differential input with $V_{\text{CM}} = 0.9\,\text{V}$; small-signal AC excitation ($+0.5/-0.5$); tail current $I_{\text{TAIL}} = 10\,\mu\text{A}$; bias voltages $V_{BN} = 0.6\,\text{V}$, $V_{BP} = 0.6\,\text{V}$; load capacitance $C_L = 1\,\text{pF}$ | Low-pass cutoff frequency ($f_c$); passband gain (dB); roll-off rate (dB/dec); stopband attenuation (dB); quality factor ($Q$); DC power consumption ($\mu\text{W}$). | Operating-point analysis to establish bias currents and DC operating conditions; AC analysis (1 Hz–10 GHz) with differential excitation to extract frequency response; cutoff frequency determined from $-3\,\text{dB}$ point; passband gain measured at low frequency; roll-off and stopband attenuation extracted from AC magnitude slope; quality factor computed from filter parameters; power calculated from supply current. |
| Folded-Cascode OTA with Integrated High-Pass Filter (HPF) | $V_{DD} = 1.8\,\text{V}$; differential input with $V_{\text{CM}} = 0.9\,\text{V}$; small-signal AC excitation ($+0.5/-0.5$); tail current $I_{\text{TAIL}} = 10\,\mu\text{A}$; bias voltages $V_{BN} = 0.7\,\text{V}$, $V_{BP} = 1.0\,\text{V}$; load capacitance $C_L = 1\,\text{pF}$ | High-pass cutoff frequency ($f_c$); passband gain (dB); roll-off rate (dB/dec); stopband attenuation (dB); quality factor ($Q$); DC power consumption ($\mu\text{W}$). | Operating-point analysis to establish bias currents and DC operating conditions; AC analysis (1 Hz–10 GHz) with differential excitation to extract frequency response; cutoff frequency determined from $-3\,\text{dB}$ point; passband gain measured at high frequency; roll-off and stopband attenuation extracted from AC magnitude slope; quality factor computed from filter parameters; power calculated from supply current. |
| Folded-Cascode OTA with Integrated Band-Pass Filter (BPF) | $V_{DD} = 1.8\,\text{V}$; differential input with $V_{\text{CM}} = 0.9\,\text{V}$; small-signal AC excitation ($+0.5/-0.5$); tail current $I_{\text{TAIL}} = 10\,\mu\text{A}$; bias voltages $V_{BN} = 0.7\,\text{V}$, $V_{BP} = 1.0\,\text{V}$; load capacitance $C_L = 1\,\text{pF}$ | Center frequency ($f_0$); bandwidth (Hz); peak gain (dB); quality factor ($Q$); DC power consumption ($\mu\text{W}$). | Operating-point analysis to establish bias currents and DC operating conditions; AC analysis (1 Hz–10 GHz) with differential excitation to extract band-pass frequency response; center frequency identified at peak gain; bandwidth measured between $-3\,\text{dB}$ points; quality factor computed as $Q = f_0/\text{BW}$; peak gain extracted from AC magnitude response; power calculated from supply current; |

# B. Prompts used of AUTOSIZER

## B.1. Circuit Understanding Prompt

**Circuit Understanding Prompt**

```
You are an expert analog circuit designer. Your task is to understand the circuit
topology and the role of each component, with focus on the optimization variables.

**Circuit Name:** {subckt_name}
**Circuit Netlist:**
```
{ota_subckt_template}
```
**Fixed Design Parameters (DO NOT optimize):**
{params}

**Optimization Variables (WILL be optimized):**
{variables}

**Testbench:**
```
{testbench_template}
```
**Performance Metrics:**
The following metrics will be evaluated:
{metrics_list}

## Your Task - Part 1: Circuit Analysis

**IMPORTANT: Keep each main section to 3-5 sentences total. Focus on how the
OPTIMIZATION VARIABLES affect circuit performance, not the fixed parameters.**

### 1. Circuit Topology Overview (3-5 sentences total)
Identify the circuit type from the netlist, describe the overall architecture, and
explain the signal flow from input to output.

### 2. Optimization Variables Mapping (3-5 sentences total)
For each optimization variable, identify which transistors it controls (from the netlist
and scaling rules). Group the variables by the functional role of transistors they
control.

### 3. Optimization Variables Impact on Performance (3-5 sentences total for each
sub-section)

Impact on {metric_name}:
Explain how the optimization variables {{variables}} affect {metric_key}.
Focus on which variables have the strongest impact on {metric_key} and why.

### 4. Variable Interactions (3-5 sentences total)
Describe any interactions between optimization variables: which must be scaled together
for matching, which have conflicting effects, and which have synergistic effects.

### 5. Key Insights for Optimization (3-5 bullet points, 1 sentence each)
Provide 3-5 critical insights about optimizing this circuit, focusing on the
optimization variables.

## Output Requirements

You must provide your analysis in **valid JSON format only**.

Your response should contain ONLY the JSON object with no additional text before
or after.
```

```
### JSON Structure
```json
{{
  "circuit_topology_overview": "3-5 sentences identifying the circuit type, describing
  the overall architecture, and explaining signal flow from input to output.",

  "optimization_variables_mapping": "3-5 sentences for each optimization variable,
  identifying which transistors it controls and grouping variables by functional role.",

  "optimization_variables_impact": {impact_json_str},

  "variable_interactions": "3-5 sentences describing interactions between optimization
  variables: which must be scaled together for matching, which have conflicting effects,
  and which have synergistic effects.",

  "key_insights_for_optimization": [
    "First critical insight about optimizing this circuit (1 sentence)",
    "Second critical insight about optimizing this circuit (1 sentence)",
    "Third critical insight about optimizing this circuit (1 sentence)",
    "Fourth critical insight (optional, 1 sentence)",
    "Fifth critical insight (optional, 1 sentence)"
  ]
}}
```

**CRITICAL REQUIREMENTS:**
1. Your entire response MUST be valid JSON – no markdown, no explanations, no text
outside the JSON object
2. Do NOT wrap the JSON in code blocks or backticks
3. Each section should be 3-5 sentences focused on OPTIMIZATION VARIABLES
4. The "key_insights_for_optimization" array must contain 3-5 strings (bullet points
as array elements)
5. All strings must properly escape special characters (quotes, newlines, etc.)

Focus your analysis on the OPTIMIZATION VARIABLES and their impact on performance.
```

## B.2. Initial Search Space Construction Prompt

**Initial Search Space Construction Prompt**

```
You are an expert analog circuit optimizer. Based on the circuit understanding analysis,
your task is to rank all optimization variables by their impact on the target metric
and determine the search space.

**Circuit Name:** {subckt_name}
**Optimization Target Metric:** {target_metric}
**Number of Variables to Actively Optimize:** {num_variables_to_optimize}
**Total Number of Variables:** {total_num_variables}

**Available Discrete Values:**
{variable_ranges}

**Scaling Rules (if applicable):**
{scaling_rules}

**Circuit Understanding Analysis:**

**Variable Impact on Metrics:**
{variable_impact_summary}

**Variable Interactions:**
{variable_interactions}
```

```
**Key Optimization Insights:**
{key_insights}

## Your Task - Part 2: Variable Prioritization and Search Space Configuration

Your task is to:

1. **Rank ALL optimization variables** (1 to {total_num_variables}) based on their
impact on {target_metric}
2. **Select the top {num_variables_to_optimize} variables** to actively optimize
3. **For the top {num_variables_to_optimize} variables**: Provide sparse search ranges
covering small to large values (3-7 values each, including extremes)
4. **For the remaining variables**: Determine appropriate fixed values
(will NOT be optimized)

**Important Considerations:**
- Prioritize variables that have the strongest impact on {target_metric}
- Consider matching requirements and circuit topology
- Balance between search space size and coverage
- Fixed values should enable good performance while allowing optimized variables to
have maximum impact

### JSON Structure
```json
{{
  "optimization_target": "copy target metric here",
  "num_variables_to_optimize": {num_variables_to_optimize},

  "variable_ranking": [
    {{
      "rank": <integer>,
      "variable": "<variable_name>",
      "impact_on_target": "<critical/high/medium/low>",
      "reasoning": "<CONCISE explanation max 100 chars - no newlines>"
    }}
  ],

  "optimization_configuration": {{
    "variables_to_optimize": {{
      "<variable_name>": {{
        "rank": <integer>,
        "search_space": [<value1>, <value2>, <value3>, ...],
        "num_choices": <integer>,
        "range_reasoning": "<CONCISE max 100 chars - no newlines>",
        "expected_behavior": "<CONCISE max 80 chars - no newlines>",
        "sensitivity": "<high/medium/low>"
      }}
    }},

    "variables_fixed": {{
      "<variable_name>": {{
        "rank": <integer>,
        "fixed_value": <numeric_value>,
        "fixed_reasoning": "<CONCISE max 100 chars - no newlines>",
        "why_this_value": "<CONCISE max 80 chars - no newlines>",
        "risk_if_suboptimal": "<low/medium/high>"
      }}
    }}
  }},

  "search_space_summary": {{
    "original_full_space": <integer>,
```

```
      "reduced_search_space": <integer>,
      "reduction_factor": <number>,
      "calculation": "<mathematical expression>",
      "explanation": "<CONCISE max 100 chars – no newlines>"
   }}
}}
```

## B.3. Optimization Algorithm Orchestration Prompt

**Optimization Algorithm Orchestration Prompt**

```
ADVANCED SEARCH METHODS AVAILABLE
### Method Arsenal:

**1. 'lhs' (Latin Hypercube Sampling)** – Pure Exploration
– **Use when**: Early exploration, design space poorly understood, <10 previous designs
– **Sample size**: Based on search space – larger space needs more samples (15-30%
of space, min 15)
– **Parameters**: 'seed' (change each iteration for diversity)

**2. 'genetic' (Genetic Algorithm)** – Evolutionary Exploration
– **Use when**: Want diverse solutions, need robust global search, 10-50 previous
designs
– **Sample size**: Based on search space – scales with complexity (10-20% of space,
20-50 samples typical, larger for big spaces)
– **Key Parameters**:
  – 'mutation_rate': 0.2 (normal) → 0.4+ (stuck)
  – 'crossover_rate': 0.8 (default)
  – 'tournament_size': 3

**3. 'bayesian' (Gaussian Process Bayesian)** – Intelligent Exploitation
– **Use when**: 25+ previous results, need sample efficiency, discrete space
performance uncertain
– **Sample size**: Based on search space – more samples for larger spaces
(5-15% of space, 10-20 samples typical, can go higher for large spaces)
– **Key Parameters**:
  – 'acquisition_function': "EI" (balanced), "LCB" (explore), "UCB" (stuck),
  "PI" (refine)
  – 'exploration_weight': 0.1-0.3 for EI/PI, 2.0-3.5 for LCB/UCB

**4. 'adaptive' (Multi-Strategy Balance)** – Balanced
– **Use when**: Unsure whether to explore/exploit, want safe balanced approach
– **Sample size**: Based on search space – moderately scales with size
(10-20% of space, 15-25 samples typical, adjust for large spaces)
– **Key Parameters**:
  – 'explore_weight': 0.5 (default), 0.6+ (low diversity)
  – 'exploit_weight': 0.5 (default), 0.6+ (late stage)
  – 'random_weight': 0.2 (default), 0.3 (stuck)

**5. 'annealing' (Simulated Annealing)** – Escape Local Optima
– **Use when**: Stuck in plateau, suspect local optimum, need to escape
– **Sample size**: Based on search space – needs more samples for larger spaces
(8-15% of space, 12-20 samples typical, scale up for big spaces)
– **Key Parameters**:
  – 'initial_temperature': 2.0 (default), 3.0 (desperate)
  – 'cooling_rate': 0.95

**6. 'multistart' (Multi-Start Local Search)** – Find Alternatives
– **Use when**: Verify convergence, find diverse good designs, late stage
– **Sample size**: Based on search space – more starts for larger spaces
(8-15% of space, 12-25 samples typical, proportional to space complexity)
```

```
- **Key Parameters**:
  - `n_starts`: 5 (default), 7+ (want diversity or large space)
  - `search_radius`: 2

## STRATEGIC DECISION FRAMEWORK

**Diagnose Current Situation:**
- Exploring? → LHS, Genetic, Adaptive
- Exploiting? → Bayesian, Optuna
- Stuck? → Annealing, Multistart
- Converged? → Stop

**Key Factors to Consider:**
1. **Data availability**: <10 designs → LHS/Genetic; 15-25 → Bayesian 30+ → Any method
2. **Improvement trend**: >5% → keep exploring; 1-5% → balance; <1% → exploit or stop
3. **Plateau detection**: 2 iterations no improvement → switch method;
3+ → consider stopping
4. **Search space size**: ALL methods scale with space - larger spaces need more samples

**Parameter Tuning Rules:**
- Stuck/identical designs → increase exploration (higher mutation_rate,
exploration_weight, temperature)
- Low diversity (std < 0.01) → increase randomness
- Steady improvement → continue current approach
- Approaching convergence → increase exploitation

## STOPPING RULES
**STOP if:**
1. User specification is met (PRIMARY GOAL)
2. <2% improvement over 2 iterations with diverse methods
3. 3+ iteration plateau with method diversity

**Sample Size Guidelines:**
- **All methods scale with search space size**
- **Exploration methods** (LHS, Genetic, Adaptive): 15-30% of space
- **Exploitation methods** (Bayesian, Optuna, Annealing, Multistart): 5-15% of space
- **LLM Direct**: 5-15% of space (cap individual batches at 20-30 for token limits)
- **General rule**: Larger search space → more samples needed for effective coverage
- **Always**: Cap at remaining budget, consider iteration efficiency

## RESPONSE FORMAT
**CRITICAL - READ THIS CAREFULLY**

### TEMPLATE 1: If you want to CONTINUE optimizing (use "search")
```json
{
  "action": "search",
  "method": "PUT_ALGORITHM_NAME_HERE",
  "n_samples": PUT_NUMBER_HERE,
  "parameters": {
    // Include only relevant parameters for the chosen method
    // For example, if using genetic:
    "mutation_rate": 0.2,
    "crossover_rate": 0.8,
    "tournament_size": 3
    // See method descriptions above for parameter details
  },
  "reasoning": "YOUR_REASONING_HERE",
  "confidence": "high or medium or low",
  "expected_improvement": "YOUR_EXPECTED_IMPROVEMENT",
  "convergence_assessment": "YOUR_CONVERGENCE_ANALYSIS"
}
```

```
### TEMPLATE 2: If you want to STOP optimizing (use "stop")
```json
{
  "action": "stop",
  "reasoning": "WHY_STOPPING",
  "confidence": "high or medium or low",
  "expected_improvement": "N/A – converged",
  "convergence_assessment": "CONVERGENCE_EVIDENCE"
}
```

## B.4. Search Space Regeneration Prompt

**Search Space Regeneration Prompt**

```
"""You are an expert optimization advisor. RE-EVALUATE and REGENERATE the search space
based on optimization results.

## CONTEXT
After {iterations_completed} iterations with {total_designs} designs evaluated,
re-analyze the circuit and adapt the search strategy.

## CIRCUIT NETLIST (Re-analyze based on actual results)
```
{netlist}
```

**Task:** Re-analyze this circuit considering the optimization results below.
Your previous understanding may have been incorrect.

## ORIGINAL SEARCH SPACE (Your Constraints)

### Available Optimization Variables
**You can ONLY optimize these variables (from YAML config):**
{available_variables}

### Fixed Parameters (CANNOT be changed to variables)
**CRITICAL: These parameters are PERMANENTLY FIXED and CANNOT be optimized
or unfixed:**
{fixed_parameters}

**WARNING: DO NOT use any variable names from the "Fixed Parameters" list above in
your optimization_configuration. They are not available for optimization under
any circumstances.**

### Value Ranges (Your choices MUST be subsets of these)
{value_ranges}

**Original Full Search Space:** {original_search_space_size} combinations

## CURRENT CONFIGURATION

### Search Space Comparison
{search_space_comparison}

### Optimized Variables
{optimized_vars_section}

### Fixed Variables
{fixed_vars_section}

**Current Search Space:** {current_search_space} combinations
```

```
## OPTIMIZATION RESULTS

### Performance Progression
{progression_section}

### Convergence Status
- Status: {convergence_status}
- Assessment: {convergence_reason}
- Best FOM: {best_fom}
- Stagnant: {stagnant}

### Variable Impact
{impact_section}

### Issues Detected
{issues_section}

### Top Designs
{top_designs_section}

## YOUR TASK
Decide on action based on the results:
- `continue_current`: Keep current configuration (if performing well and not stagnant)
- `expand_ranges`: Expand variable ranges (preferred if ANY designs at boundaries)
- `narrow_ranges`: Narrow to promising regions (only if very clear winner region)
- `unfix_variables`: Unfix some fixed variables (preferred if stagnation or suboptimal)
- `change_focus`: Swap optimized/fixed variables (if optimized var has low impact)
- `converged`: Optimization complete (only if target met or exhausted search)

## DECISION GUIDELINES

**Expand ranges (PREFERRED ACTION):**
- 20%+ of top designs at boundaries (lowered threshold)
- Current range seems restrictive
- FOM improvements still occurring
- Action: Add 2-3 adjacent values from original ranges on each boundary
- Bias: Favor expansion over narrowing

**Unfix variables (HIGHLY ENCOURAGED):**
- Any fixed variable could impact performance
- Stagnation detected (no improvement for 2+ iterations)
- Current optimized variables explored adequately
- Suspect interactions between fixed and optimized variables
- Action: Unfix 1-2 most promising fixed variables with 5-7 values
- Priority: Unfix before declaring convergence

**Narrow ranges (USE SPARINGLY):**
- Top 80%+ designs cluster tightly in middle 30% of range
- Very clear single winner region (not just top design)
- Action: Focus on 4-6 best values only
- Warning: Only use if extremely confident

**Change focus (ENCOURAGED FOR EXPLORATION):**
- Optimized variable shows minimal variance in top designs (>70% same value)
- All top designs converged to 1-2 values for a variable
- Action: Fix the converged variable, unfix a new one
- Enables exploring new dimensions

**Continue current:**
- Making steady progress (improvement in last iteration)
- Haven't explored current space adequately (< 40% of combinations)
- No clear issues with current configuration
- Use temporarily, but bias toward taking action
```

```
**Converged (LAST RESORT):**
- Best FOM exceeds or meets target specification
- Improvements < 1% for 4+ consecutive iterations
- All promising variables explored with expanded ranges
- Fixed variables already unfixed and tested
- Warning: Only declare convergence after aggressive exploration

## DECISION PRIORITY ORDER
1. **Unfix variables** - if any fixed variables remain and performance stagnating
2. **Expand ranges** - if any boundary clustering or improvements still occurring
3. **Change focus** - if optimized variable converged but performance suboptimal
4. **Continue current** - if making progress and space not fully explored
5. **Narrow ranges** - only if overwhelming evidence of tight optimal region
6. **Converged** - only after exhausting all exploration options

## CRITICAL CONSTRAINTS
1. **ONLY use variables from "Available Optimization Variables" section above**
2. **Value ranges MUST be subsets of the ranges shown in "Value Ranges" section**
3. **DO NOT create new variables like 'L' if they are fixed parameters**
4. **DO NOT use values outside the allowed ranges**
6. **Each variable: 5-7 discrete values (increased from 3-7 for more exploration)**
7. **Variables in "Fixed Parameters" section CAN be unfixed and optimized**
8. **Bias: When uncertain, choose expansion or unfixing over narrowing or convergence**

## EXPLORATION PHILOSOPHY
- **Default to action**: Prefer exploring (expand/unfix) over staying static
- **Maximize search space**: Unfix variables aggressively before claiming convergence
- **Trust the data**: Only narrow when >80% of evidence supports it
- **Avoid premature convergence**: If in doubt, explore more

## OUTPUT (JSON ONLY)

{{{{
  "optimization_target": "{target_metric}",
  "regeneration_reasoning": "<why regenerating>",
  "action_taken": "<action>",
  "changes_from_previous": "<summary of changes>",

  "variable_ranking": [
    {{{{
      "rank": <1-n>,
      "variable": "<name>",
      "impact_on_target": "<critical|high|medium|low>",
      "reasoning": "<explanation>"
    }}}}
  ],

  "optimization_configuration": {{{{
    "variables_to_optimize": {{{{
      "<variable_name>": {{{{
        "rank": <rank>,
        "search_space": [<values>],
        "num_choices": <number>,
        "range_reasoning": "<why this range>",
        "expected_behavior": "<how target changes>",
        "sensitivity": "<high|medium|low>",
        "change_from_previous": "<what changed>"
      }}}}
    }}}},

    "variables_fixed": {{{{
      "<variable_name>": {{{{
        "rank": <rank>,
        "fixed_value": <value>,
```

```
        "fixed_reasoning": "<why fixed>",
        "why_this_value": "<why this value>",
        "risk_if_suboptimal": "<risk>",
        "change_from_previous": "<what changed>"
      }}}}
    }}}}
  }}}},

  "search_space_summary": {{{{
    "original_full_space": <total>,
    "reduced_search_space": <total>,
    "reduction_factor": "<factor>",
    "change_factor": "<expansion or reduction>",
    "calculation": "<calculation>",
    "explanation": "<explanation>"
  }}}},

  "expected_improvement": "<expected gain>",
  "confidence": "<high|medium|low>"
}}}}

Return ONLY JSON based on data.
"""
```

