# OpenReview forum: "AutoSizer: Automatic Sizing of Analog and Mixed-Signal Circuits via Large Language Model (LLM) Agents"
_ICML.cc/2026/Conference — ICML 2026 regular_

### Official Review · Reviewer_aqKx · 2026-03-05

**Soundness:** 2
**Presentation:** 3
**Significance:** 3
**Originality:** 2
**Overall Recommendation:** 4
**Confidence:** 4

**Summary:**

The paper proposes AutoSizer, a multi-agent LLM framework for automatic sizing of analog and mixed-signal circuits that combines circuit understanding, adaptive construction of the search space, and numerical optimization within a two-loop optimization workflow. The outer loop analyzes optimization history to revise variable priorities and parameter ranges, while the inner loop performs simulation-driven optimization using standard algorithms (e.g., genetic algorithms and Bayesian optimization). The paper also introduces AMS-SizingBench, a benchmark of 24 circuits across multiple difficulty levels built on the open SkyWater 130nm PDK, and reports improved solution quality and convergence compared with traditional optimizers and prior LLM-based sizing agents.

**Compliance With Llm Reviewing Policy:**

Affirmed.

**Final Justification:**

The authors provided additional experiments and extensive clarifications that successfully addressed my concerns throughout the rebuttal process as well as providing anonymous access to the AutoSizer codebase. These updates significantly strengthen the paper's claims, justifying an upgrade of my recommendation to 4.

**Key Questions For Authors:**

Q1 - Computational and API Cost.
Could the authors report the approximate LLM token usage and the monetary cost per circuit-optimization run?

Q2 - Adaptation Dynamics of the Outer Loop.
Could the authors provide statistics on how frequently the outer loop modifies the search space (e.g., number of iterations before variable expansion or re-prioritization)?

Q3 - Validation of the FOM.
Could the author add clarifications on FOM's individual metrics values? That would improve the interpretability of the reported results. Moreover, including representative simulation plots (e.g., key performance metrics such as gain, bandwidth, or transient responses) for selected circuits would strengthen empirical validation and better demonstrate that the resulting designs meet practical design requirements in realistic EDA workflows.

Q4 - Open-Source Release.
Do the authors plan to release AutoSizer and/or AMS-SizingBench as open-source resources? If so, could they clarify the release timeline and scope (e.g., code, prompts, evaluation scripts, and datasets)? If not, how do they envision enabling reproducibility and broader adoption of the proposed framework?

Clarifications addressing the behavior of the adaptive outer loop, the individual metric values, and the availability of benchmark resources will improve the assessment of the paper.

**Limitations:**

Yes.

**Strengths And Weaknesses:**

Strengths
S1 - Prioritization Strategy.
The proposed approach to identifying high-impact parameters and narrowing the search space is well motivated and practically meaningful, potentially reducing optimization costs in high-dimensional design spaces.

S2 - Diverse Benchmark Circuits.
The introduction of AMS-SizingBench with multiple circuit types enhances empirical validation and broadens the range of evaluation scenarios.

S3 - Two-Loop Meta-Optimization Architecture.
The framework separates numerical optimization from adaptive problem formulation. The outer loop analyzes optimization outcomes (e.g., convergence behavior) and dynamically revises the search space by re-prioritizing variables or adjusting parameter ranges, enabling progressive refinement.

Weaknesses
W1 - Limited Analysis of Search-Space Adaptation Behavior.
While the framework adaptively revises variable priorities and parameter ranges during optimization, the paper does not provide quantitative statistics describing how often the outer loop revises the search space or how these revisions correlate with optimization improvements across benchmark circuits. Additional diagnostics or visualizations of these adaptation dynamics would help assess the effectiveness of the outer-loop refinement process.

W2 - Missing Anonymous Access to Benchmark Resources.
Although the paper states that AMS-SizingBench is open source, the submission does not provide an anonymous repository or any supplementary material during review. Providing such resources would improve reproducibility and allow reviewers to verify the benchmark design and evaluation pipeline.

W3 - Limited Clarity on FOM.
Table 2 reports a Figure of Merit (FOM) used to compare optimization outcomes across circuits. However, the paper does not clearly report the values of the individual FOM metrics across different circuit types and specifications. While the FOM components are defined in Equation 1 and individual circuit targets are listed in the Appendix, Table 2 would benefit from a summary of which specific performance metrics (e.g., gain, bandwidth, power) were the primary drivers of the improved FOM for each circuit.

---

> ### Author Rebuttal · Authors · 2026-03-31
>
> We appreciate the reviewer's careful comments and provide our responses below.
>
> >**[W1]** Limited Analysis of Search-Space Adaptation Behavior.
> >
>
> We address all sub-questions raised: what signals trigger each transition (§1), how frequently the outer loop revises the search space, how many iterations elapse before each revision, and how these revisions correlate with FOM improvement (§2).
>
> ### 1. What Triggers Search Space Transitions
>
> At each regeneration cycle boundary, the LLM receives the CIRCUIT_REUNDERSTANDING_PROMPT (See Appendix B4) containing five categories of feedback signals programmatically extracted from the inner optimization loop — no manual curation or hard-coded heuristics are involved:
>
> | Signal | What the LLM Sees | Example Trigger |
> |:-------|:-------------------|:----------------|
> | Performance progression | FOM history across all iterations (e.g., 0.08 → 0.10 → 0.105 → 0.105) | Stagnation triggers expand_ranges or unfix_variables |
> | Convergence status | Whether FOM improved in the last 2 iterations, with stagnation flag | stagnant=True strongly encourages variable unfixing |
> | Variable impact analysis | Which optimized variables show variance vs. convergence in top designs | If 70%+ of top designs converge to one value → fix it, unfix another |
> | Issue detected | Whether top designs cluster at edges of search ranges | 20%+ at boundaries → expand_ranges to explore beyond current limits |
> | Top design details | Full parameter values and metrics of best N designs | Reveals which variables are truly driving performance |
>
> Based on these signals, the LLM selects one of six actions: continue_current, expand_ranges, narrow_ranges unfix_variables, change_focus, or converged.
>
> ### 2. Revision Frequency, Iteration Count, and FOM Sensitivity
>
> | Difficulty | Outer-Loop Action | Frequency | Avg FOM Δ | Iterations Before Action (mean±std) |
> |---|---|---|---|---|
> | Easy | expand_ranges / unfix_variables | 36.4% | +1.5% | 104±41 |
> | | narrow_ranges | 6.0% | +0.5% | 164±32 |
> | | change_focus | 7.1% | +0.3% | 154±63 |
> | | continue_current | 50.5% | +0.0% | 125±64 |
> | Medium | expand_ranges / unfix_variables | 52.4% | +2.5% | 104±80 |
> | | narrow_ranges | 20.7% | +1.3% | 118±21 |
> | | change_focus | 16.3% | +0.3% | 155±36 |
> | | continue_current | 10.6% | +0.1% | 191±80 |
> | Hard | expand_ranges / unfix_variables | 57.1% | +16.4% | 105±40 |
> | | narrow_ranges | 10.0% | -3.8% | 180±48 |
> | | change_focus | 20.0% | +2.9% | 183±52 |
> | | continue_current | 12.9% | +1.6% | 197±35 |
>
> On easy circuits, continue_current dominates, confirming the inner loop alone suffices for low-dimensional problems. On hard circuits, expand_ranges/unfix_variables grows to 57.1% (+16.4% FOM), showing the LLM increasingly relies on search-space reshaping. The action timing reveals a principled ordering: expansion occurs early (~104 iterations), while narrow_ranges and change_focus appear later (154–197 iterations, requiring 1.5–2 cycles of evidence), and continue_current appears latest on hard circuits (197±35). This expand-early, narrow-late, continue-only-with-evidence strategy emerges from optimization feedback, not hard-coded rules.
>
> >**[W2&Q3]** Missing Anonymous Access to Benchmark Resources.
> >
> We will release the full AutoSizer codebase, AMS-SizingBench (including all circuit netlists, simulation configurations, and prompts) after the review process concludes.
>
> >**[W3&Q2]** Limited Clarity on FOM.
> >
> The table below decomposes FOM into constituent metrics across optimization cycles for four circuit groups.
>
> | Circuit Type| Metric | Dir | Cycle 0 | Cycle 1 | Cycle 2 | Δ(0→final) |
> |---|---|:---:|---:|---:|---:|---:|
> | OTAs | DC Gain (dB) | ↑ | 27.63 | 34.41 | 35.00 | +26.7% |
> | | UGBW (MHz) | ↑ | 26.24 | 250.42 | 235.71 | +798.4% |
> | | Power (µW) | ↓ | 294.2 | 280.0 | 274.2 | -6.8% |
> | | Avg FOM | ↑ | 1.017 | 2.459 | 2.498 | +145.6% |
> | Mixed-Signal | Gain (dB) | ↑ | 15.69 | 16.34 | 18.42 | +17.4% |
> | | UGBW (MHz) | ↑ | 125.9 | 125.9 | 131.8 | +4.7% |
> | | THD (dB) | ↓ | -8.70 | -18.10 | -27.30 | +213.8% |
> | | Power (µW) | ↓ | 1382.5 | 1191.3 | 1060.1 | -23.3% |
> | | Avg FOM | ↑ | 1.060 | 2.666 | 5.205 | +390.8% |
> | Voltage Ref | PSRR (dB) | ↑ | 50.46 | 51.09 | 50.62 | +0.3% |
> | | Power (µW) | ↓ | 9390 | 9380 | 9348 | -0.4% |
> | | Line Reg | ↓ | 1.69 | 1.69 | 1.62 | -4.1% |
> | | Avg FOM | ↑ | 89.85 | 90.35 | 92.7 | +3.2% |
> | Digital Gates | Delay (ps) | ↓ | 3976 | 3995 | 3962 | -0.4% |
> | | Energy (fJ) | ↓ | 30.27 | 29.42 | 28.35 | -6.3% |
> | | Power (µW) | ↓ | 3.03 | 2.94 | 2.84 | -6.3% |
> | | Avg FOM | ↑ | 0.249 | 0.230 | 0.250 | +0.4% |
>
> Amplifiers show massive UGBW gains as the primary FOM driver. Mixed-Signal improvement is dominated by THD and Power. Voltage References achieve balanced improvement across PSRR and line regulation. Digital Gates show modest FOM change as energy and delay tradeoffs largely offset.
>
> >**[Q1]** Computational and API Cost.
> >
> See Reviewer **inKn** **[W2]**

---

> > ### Author Rebuttal · Reviewer_aqKx · 2026-04-02
> >
> > The rebuttal provides helpful additional details and clarifications, particularly regarding the triggers for search-space adaptation (W1) and the quantitative breakdown of FOM improvements across different circuit categories (W3). However, concerns regarding empirical depth remain.
> >
> > **Validation via Simulation Waveforms** The FOM decomposition in the rebuttal clarifies which metrics drove the numerical gains. However, for an EDA-focused contribution, numerical scores are an abstraction. The absence of representative simulation plots (e.g., gain-phase Bode plots for stability or transient responses for settling behavior) remains a weakness. These plots are standard in the field to demonstrate that the designs produced by the agent are physically sound and meet practical design requirements beyond simple constraint-satisfaction metrics.
> >
> > Morevoer, while the Appendix provides the prompts and circuit configurations and the authors commit to release the full codebase after the review process, the absence of an anonymous code repository or the underlying benchmark execution environment during the review process limits the ability to verify the framework’s "end-to-end" integration.

---

> > > ### Author Response · Authors · 2026-04-04
> > >
> > > We sincerely thank the reviewer for the thorough and constructive feedback. We greatly appreciate the time and effort spent in evaluating our work, and we are grateful for the specific suggestions that have helped us strengthen the paper.
> > >
> > > ### Validation via Simulation Waveforms
> > >
> > > We fully agree with the reviewer that waveform-level validation is the standard practice in the EDA community and is essential for demonstrating physical soundness. Following the reviewer's valuable suggestion, we have added 7 representative simulation plots (available in the `Figures/` folder of our anonymous repository [[LINK](https://anonymous.4open.science/r/AutoSizer-ICML-29196)]) generated from AutoSizer's optimized designs across multiple circuit categories:
> > >
> > > - **Gain-phase Bode plot (indirect-compensated amplifier):**
> > >   DC gain = 68.2 dB, UGBW = 0.3 MHz, phase margin = 65.4°, confirming high-gain multi-stage amplifier behavior with adequate stability margin.
> > >
> > > - **LDO load-step transient (2-stage LDO regulator):**
> > >   Under a 1→10 mA load step, the optimized design exhibits 15.5 mV droop, 5.6 mV overshoot, and settles within ±0.1% in 180.8 µs, meeting practical regulation requirements beyond simple constraint satisfaction.
> > >
> > > - **Bandgap reference vs. temperature (true bandgap):**
> > >   Vref = 1.219 V at 27°C with only 4.89 mV variation across −40°C to 125°C (TC = 24.3 ppm/°C), and line regulation of 0.25%/V from 1.8–3.3 V supply, demonstrating that AutoSizer can size a hierarchical analog system to achieve near-ideal silicon-bandgap behavior.
> > >
> > > - **Ring oscillator transient:**
> > >   Clean 3-stage oscillation at 2.48 GHz with correct 120° phase offsets between stages.
> > >
> > > - **VCO tuning curve:**
> > >   Monotonic frequency-vs-control characteristic from 1342–1982 MHz (38.5% tuning range), suitable for PLL integration.
> > >
> > > - **Inverter transient:**
> > >   Rail-to-rail switching with tpHL = 10.9 ps, tpLH = 22.4 ps (avg 16.7 ps), confirming correct digital functionality.
> > >
> > > - **Band-pass filter Bode plot:**
> > >   Near-unity passband gain (0.1 dB) with cutoff at 41.2 kHz, showing expected bandpass frequency response.
> > >
> > > We hope these results, spanning amplifiers, regulators, references, oscillators, and filters, help demonstrate that AutoSizer produces waveform-level behavior consistent with what an experienced analog designer would expect, not merely numerically compliant designs. We thank the reviewer for pushing us to include this important evidence.
> > >
> > > ### Anonymous Code Repository
> > >
> > > We sincerely appreciate the reviewer’s emphasis on enabling verification of our framework. We have released an anonymous repository at [[LINK](https://anonymous.4open.science/r/AutoSizer-ICML-29196)] containing the full AutoSizer framework, including the AMS-SizingBench benchmark (24 circuit configurations), all dedicated SPICE simulation scripts, benchmark execution environment (via Docker container), and the LLM-guided optimization loop. We hope this fully addresses the reviewer's concern.

---

### Official Review · Reviewer_d4oc · 2026-03-07

**Soundness:** 2
**Presentation:** 3
**Significance:** 2
**Originality:** 2
**Overall Recommendation:** 4
**Confidence:** 3

**Summary:**

This paper presents AutoSizer, a reflective LLM-driven meta-optimization framework for analog and mixed-signal transistor sizing. It adopts a two-loop architecture where an inner loop runs conventional optimizers and an outer loop leverages LLM reasoning to iteratively refine the search space from simulation feedback. An open benchmark of 24 AMS circuits (AMS-SizingBench) is also introduced for evaluation.

**Compliance With Llm Reviewing Policy:**

Affirmed.

**Final Justification:**

Thanks for the authors' rebuttal. After considering the authors' response and other reviewers' comments, I will raise my score. However, I still have some concerns regarding the stability and interpretability of the optimization-based LLM approach. I would encourage the authors to discuss these aspects further in the revised version.

**Key Questions For Authors:**

- In Fig. 5(a), the search space undergoes notable changes across the three stages of optimization. Could the authors provide a more thorough interpretation of why the search space evolves in this particular manner? Specifically, what signals or feedback from the optimization loop trigger each transition, and how sensitive is the final performance to these search space adjustments?

**Limitations:**

Yes

**Strengths And Weaknesses:**

Strengths:
- The core idea of leveraging LLMs to constrain the design space is well-motivated and practically relevant, particularly for analog circuit optimization scenarios where designers may lack extensive experience with parameter tuning and optimization strategy selection.


- The paper is clearly written and easy to follow.


Weaknesses:
- The overall methodology essentially relies on prompt engineering to orchestrate the optimization process. It combines LHS, BO, and SA, and delegates the strategy selection among them to the LLM. However, the effectiveness of this LLM-driven strategy selection remains unconvincing. While it is true that different optimization methods excel at different design tasks, I am skeptical that an LLM can reliably choose the most suitable method for a given context, given the scarcity of analog circuit optimization data in typical LLM training corpora. No evidence or ablation study is provided to validate that the LLM's strategy choices are meaningfully better than a fixed or random schedule.


- The technical contribution is limited, as the work primarily centers on prompt engineering. Similar LLM-assisted analog sizing frameworks have already been proposed in prior work such as LEDRO and EE-Opt, and the paper does not sufficiently articulate what fundamentally distinguishes AUTOSIZER from these existing approaches beyond prompt design choices.


- As shown in Table 2, the proposed AUTOSIZER exhibits large variance across runs, which raises concerns about the robustness and reproducibility of the results. Running additional rounds could lead to substantially different conclusions, undermining the reliability of the reported comparisons.


- Minor issue: In Section 2.2 (line 158), the citation for AnalogXpert appears to be incorrect.

---

> ### Author Rebuttal · Authors · 2026-03-31
>
> We appreciate the reviewer's careful comments and provide our responses below.
>
> >**[W1]** Methodology relies on prompt engineering; LLM strategy selection is unvalidated.
>
> We appreciate this thoughtful concern. We provide two levels of evidence: (1) empirical decision logs showing adaptive behavior, (2) ablation results quantifying the benefit.
> ### 1. The LLM Makes Non-Trivial, Adaptive Decisions
>
> We logged every strategy decision the LLM made. The table below shows the complete decision trace for the bandgap circuit:
>
> | | Cycle 0 (3 vars) | Cycle 1 (5 vars) | Cycle 2 (4 vars) |
> |---|---|---|---|
> | Iter 0 | lhs n=15 | bo n=35 | bo n=25 |
> | Iter 1 | genetic n=15 | bo n=50 | bo n=35 |
> | Iter 2 | bo n=15 | lhs n=50 (re-explore) | — |
> | Iter 3 | annealing n=15 | — | — |
> | Iter 4 | lhs n=50 (re-explore) | — | — |
> | #iterations | 110 | 135 | 60 |
> | Vars optimized | 3 (L_ra, L_rb, L_mp) | 5 (W_mp, L_mp, W_mn, L_ra, L_rb) | 4 (W_mp, W_mn, L_mn, L_rb) |
> | Best FOM | 0.1085 | 0.1152 (+6.2%) | **0.1209 (+11.5%)** |
>
> We summarize the three patterns emerge in our benchmark that **no fixed or random schedule can replicate**:
>
> | Adaptive behavior | What the LLM did | Why fixed/random fails |
> |---|---|---|
> | Mid-cycle re-exploration | Detected stagnation after bo and switched back to LHS | Fixed never re-explores; random re-explores arbitrarily |
> | Circuit-specific selection | Bandgap: 4 methods incl. SA. Switched-cap: skipped SA, alternated BO↔LHS | One fixed schedule cannot fit different FOM landscapes |
> | Space aware sizing | n=15 for 3 vars → n=35–50 for 5 vars → n=25–35 for 4 vars | Fixed uses constant n regardless of space size |
>
> The LLM receives structured feedback after each iteration: current search space dimensions and remaining budget, per-method FOM improvement history, parameter convergence statistics. This enables it to select suitable method.
>
> ### 2. Ablation Study: Fixed vs. Random vs. LLM Schedule
>
> To directly quantify the benefit, we compared three strategy selection modes under the same total budget (300 evaluations, 3 cycles):  Fixed schedule follows a predetermined explore to exploit pattern (LHS then Bayesian with decreasing n). Random schedule draws method and sample size uniformly at each iteration. AutoSizer consistently outperforms both fixed and random schedule baseline.
>
> | Circuit | Fixed Schedule | Random Schedule | AutoSizer (LLM) |
> |:----------------|:--------------:|:---------------:|:----------------:|
> | bandgap | 0.08 | 0.06 | **0.12** |
> | five_trans_ota | 0.58 | 0.45 | **0.61** |
> | folded_cascode | 1.33 | 1.05 | **5.20** |
> | telescopic_ota | 0.05 | 0.04 | **0.10** |
>
>
> >**[W2]** The technical contribution is limited, as the work primarily centers on prompt engineering.
> >
> The distinctions from prior work are fundamental: LEDRO reduces the search space once with no feedback loop; ADO-LLM uses in-context learning for BO but lacks self-reflection; EE-Sizer generates designs directly without explicit optimization algorithms. AutoSizer's contributions are **algorithmic**, not prompt engineering: **(1) adaptive method selection**, the LLM selects and parameterizes search methods based on convergence signals each iteration, outperforming fixed and random schedules in ablation; **(2) outer-loop search space revision** the LLM revises variable priorities based on simulation results.
>
>
> >**[W3]** As shown in Table 2, the proposed AUTOSIZER exhibits large variance across runs.
> >
>
> We respectfully point out that AutoSizer's variance is comparable to or lower than all baselines. For instance, AutoSizer achieves ± 3.4 std — the lowest among all methods (LEDRO: ± 8.5, ADO-LLM: ± 9.1, EE-Sizer: ± 11.5, TuRBO: ± 12.7). Where higher variance exists, it is shared across all methods, suggesting this is an inherent characteristic of the circuit optimization landscape rather than a limitation specific to AutoSizer.
>
> >**[W4]** Incorrect citation.
> >
> We will correct the citation in the revised manuscript.
> >
> >**[Q1]** Provide a more thorough interpretation of why the search space evolves in this particular manner.
> >
> AMS optimization problems are too large to solve at once and too nonlinear to predict without data. The outer loop implements a principled decomposition: focus on the most sensitive variables first, learn from simulation results, then progressively expand based on empirical evidence. On the ring oscillator (Fig. 5a), the search space follows a focus → expand → refocus cycle, the LLM first optimizes the output stage and channel
> length (most load-sensitive), then rotates to internal stages after convergence, discovering that halving Stage 1 transistors increased frequency by 13.7% due to reduced node capacitance, yielding +51.5% FOM overall. This mirrors how experienced designers iteratively size circuits, and emerges from LLM reasoning over optimization feedback. See Reviewer **aqKx [W1]** for search space analysis.

---

> > ### Author Rebuttal · Reviewer_d4oc · 2026-04-03
> >
> > Thank you for adddressing my concerns.

---

> > > ### Author Response · Authors · 2026-04-04
> > >
> > > Thank you for taking the time to review our response. We sincerely appreciate your acknowledgment that the concerns have been fully addressed. We will incorporate the above analysis into our revised manuscript. We would be grateful if you could consider updating your score accordingly.

---

### Official Review · Reviewer_inkn · 2026-03-12

**Soundness:** 3
**Presentation:** 3
**Significance:** 4
**Originality:** 3
**Overall Recommendation:** 4
**Confidence:** 5

**Summary:**

This paper introduces AutoSizer, a multi-agent framework powered by Large Language Models (LLMs) for the automatic sizing of Analog and Mixed-Signal (AMS) circuits. The system uses a two-loop architecture: an inner loop that runs standard optimization algorithms (like Bayesian Optimization or Genetic Algorithms) to find specific device sizes, and an outer loop where the LLM reflects on the results to refine the search space and prioritize important variables. Additionally, the authors present AMS-SizingBench, an open-source benchmark with 24 diverse circuits to test such tools. Experiments show that AutoSizer finds better designs faster and more reliably than existing methods.

**Compliance With Llm Reviewing Policy:**

Affirmed.

**Final Justification:**

I decide to maintain the original score

**Key Questions For Authors:**

None

**Limitations:**

yes

**Strengths And Weaknesses:**

Strengths：
1. Innovation in Methodology: The paper introduces a "two-loop" framework that successfully combines the reasoning power of LLMs with traditional optimization algorithms. While the outer loop uses LLM agents to understand circuit structures and dynamically adjust the search space like a human expert, the inner loop handles the heavy numerical calculations.
2. Excellent Experimental Results: The authors provide a very strong evaluation by creating AMS-SizingBench, a new open-source benchmark featuring 24 diverse circuits. This is a significant improvement over previous studies that often test only a few simple designs. The experimental data clearly shows that AutoSizer is more efficient and reliable than existing tools, achieving better design results with fewer simulations.
3. Clear writing: The paper is well-written and easy to follow. The overall structure is logical, and the use of clear diagrams helps the reader quickly understand how the different agents interact within the system.

Weakness:
1. Insufficient Baseline Comparisons: There are many other space search techniques and state-of-the-art Bayesian Optimization (BO) sizing methods that were not included in the performance comparison.
2. Lack of Model Dependency Analysis: Although the results are good, the paper does not show how different LLMs (like GPT-4, Claude 3.5, or open-source Llama 3) affect the results. It is unclear if the system’s success depends too much on a specific, expensive commercial model.
3. Robustness of Prompt Engineering: The performance of the entire framework depends heavily on the design of the initial prompts. The paper lacks deep "stress tests" to show if the current prompting strategy is stable for very complex or unusual circuit designs. It is unclear whether a lot of expert work is still needed to manually tune these prompts for different cases.
4. Limited Testing on High-Dimensional Spaces: While the benchmark includes 24 circuits, the paper does not test very large systems with hundreds of variables. For such large designs, the LLM’s memory limits (context window) and reasoning errors could become serious problems.

---

> ### Author Rebuttal · Authors · 2026-03-31
>
> We appreciate the reviewer's careful comments and provide our responses below.
>
> >**[W1]** Insufficient Baseline Comparisons
>
> We added three additional optimization baselines. i.e., MCMC, UCB1 (multi-armed bandit), and SAASBO (high-dimensional Bayesian optimization). Results are averaged across circuits per difficulty level:
>
> | Method | Easy | Medium | Hard |
> |:---------|:-------------:|:-----------------------:|:-------------------:|
> | MCMC | 16.2 | 26.8 | 37.2 |
> | UCB1 | 16.8 | 26.1 | 35.8 |
> | SAASBO | 25.5 | 32.1 | 49.7 |
> | AutoSizer | **33.4** | **33.2** | **60.2** |
>
> AutoSizer outperforms all baselines across all difficulty levels, with the largest margin on Hard circuits (+21% over SAASBO).
>
> >**[W2]** Lack of Model Dependency Analysis
>
> We evaluated AutoSizer with three LLMs using identical prompts and budgets:
>
> |  | Easy |  |  | Medium |  |  | Hard |  |  |
> |:-----|:------:|:------:|:------:|:------:|:------:|:------:|:------:|:------:|:------:|
> | **LLM Model** | **FOM↑** | **Tokens (in/out)** | **Cost** | **FOM↑** | **Tokens (in/out)** | **Cost** | **FOM↑** | **Tokens (in/out)** | **Cost** |
> | Gemini 2.5 Flash | 33.4 | 85.6k/8.9k | ~$0.05 | 33.2 | 105.8k/10.9k | ~$0.06 | 60.2 | 126.5k/12.3k | ~$0.08 |
> | GPT-4o | 33.0 | 71.3k/5.0k | ~$0.22 | 33.1 | 116.5k/9.8k | ~$0.26 | 58.2 | 120.5k/12.0k | ~$0.43 |
> | Llama-3.3-70B (open) | 32.2 | 90.7k/6.5k | -- | 32.1 | 118.9k/7.8k | -- | 53.8 | 127.2k/10.6k | -- |
>
> All three LLMs achieve similar FOM, confirming that AutoSizer's performance is architecture-driven, not dependent on a specific LLM.
>
> >**[W3]** Robustness of Prompt Engineering
>
> We demonstrate prompt robustness in three ways:
>
> 1. **Same prompts across 24 diverse circuits.** Our prompts (shown in Appendix) are circuit-agnostic: the same `CIRCUIT_UNDERSTANDING_PROMPT`, `SEARCH_SPACE_REDUCTION_PROMPT`, and iteration decision prompts are used for all 24 circuits spanning inverters, OTAs, folded cascodes, bandgap references, VCOs, and switched-capacitor circuits. No per-circuit prompt tuning was performed.
> 2. **Cross-PDK generalization.** The same prompts work across SKY130 (130nm), FreePDK45 (45nm), and ASAP7 (7nm FinFET) without modification, despite fundamentally different device physics and parameterization (W/L vs. nfin).
> 3. **Cross-LLM generalization.** The same prompts produce valid optimization configurations across Gemini, GPT-4o, and Llama (W2 table above), demonstrating that the prompts capture domain knowledge in a model-agnostic way.
>
>
> >**[W4]** Limited Testing on High-Dimensional Spaces
> >
>
> We acknowledge this concern and present a concrete scaling experiment. Analog circuits inherently possess hierarchical structure, transistors within differential pairs, current mirrors, and cascode stacks are functionally equivalent and should share sizing, a standard practice among experienced engineers. Leveraging this, we added an automatic transistor grouping step in the circuit understanding phase, triggered when total variables exceed 20. The LLM analyzes the netlist and groups equivalent transistors, reducing the IAC circuit from 108 raw variables (34 transistors × 3 parameters + 6 passives) to 8 group-level variables. On this 108-variable IAC circuit, AutoSizer achieved FOM improvement from 0.101 → 6.12 (+5960%), with DC Gain improving from 38.97 to 68.16 dB. Token usage remained comparable (127.7k input / 12.8k output), confirming the approach scales without exceeding LLM context capacity.

---

> > ### Author Rebuttal · Reviewer_inkn · 2026-04-03
> >
> > Thank you for the rebuttal. I have three brief follow-up comments.
> >
> > In W3, you mention that the method also works on 45nm and 7nm FinFET technologies. I think this claim would need direct experimental evidence to be fully convincing. Different process libraries can lead to substantially different search spaces and device parameterizations, so in practice some nontrivial adaptation would likely be required.
> > The additional experiments in Q1 and Q2 are very helpful. They strengthen the paper and provide more convincing evidence for the effectiveness of the proposed framework.
> > For W4, my main suggestion is to evaluate the method on a more clearly hierarchical circuit, such as a small ADC, a nested OTA-based design, or another larger mixed-signal block. That would make the scalability claim much stronger.

---

> > > ### Author Response · Authors · 2026-04-04
> > >
> > > We sincerely thank the reviewer for the thoughtful feedback and for acknowledging the additional experiments.
> > >
> > > ### W3 — Cross-Technology Portability
> > >
> > > We appreciate the reviewer's point that different process libraries can lead to substantially different search spaces. We note that this concern was also raised by Reviewer JxhW, and we have addressed it there. However, we reiterate and further clarify our response here for completeness.
> > >
> > > To provide direct evidence, we have conducted experiments on both FreePDK 45nm and ASAP7 7nm FinFET technologies:
> > >
> > > | PDK              | Circuit            | Method    | FOM ↑ | Evals ↓ | Time (s) ↓ |
> > > |------------------|--------------------|-----------|-------|---------|------------|
> > > | FreePDK 45nm     | five_trans_ota     | LEDRO     | 2.2   | 35      | 41.5       |
> > > |                  |                    | AutoSizer | 2.9   | 21      | 32.7       |
> > > |                  | current_mirror_ota | LEDRO     | 1.6   | 230     | 256.1      |
> > > |                  |                    | AutoSizer | 2.0   | 210     | 214.5      |
> > > | ASAP7 7nm FinFET | five_trans_ota     | LEDRO     | 5.8   | 286     | 302.8      |
> > > |                  |                    | AutoSizer | 6.6   | 266     | 270.3      |
> > > |                  | current_mirror_ota | LEDRO     | 3.3   | 90      | 288.1      |
> > > |                  |                    | AutoSizer | 4.0   | 66      | 266.2      |
> > >
> > > AutoSizer consistently outperforms LEDRO across both technologies without any framework modification. This is possible because AutoSizer's architecture is fundamentally PDK-agnostic: the LLM agent's circuit understanding is grounded in circuit topology and device-level physics (e.g., "increasing W of a differential pair improves transconductance and gain"), which are universal principles across process nodes, not tied to any specific technology model parameters.
> > >
> > > The PDK-specific details (model files, device parameterizations, valid sizing ranges) are encapsulated in the YAML configuration and SPICE simulation scripts, while the LLM-guided search-space adaptation and optimization method selection are purely based on numerical simulation feedback (metric values, constraint violations, improvement trends), not on any process-specific heuristics.
> > >
> > >
> > > ### W4 — Hierarchical Circuit
> > >
> > > We thank the reviewer for this valuable suggestion. To directly address this concern, we have extended AutoSizer to a switched-capacitor low-pass filter — a nested OTA-based design as the reviewer specifically suggested.
> > >
> > > This circuit has a clear two-level hierarchy: Level 1 consists of transistor-level sizing of a two-stage OTA (NMOS differential pair, PMOS active load, output stage, bias circuit), while Level 2 is the SC filter system built around this OTA . AutoSizer optimizes 5 transistor-level variables, and these propagate through multiple abstraction layers to affect system-level metrics.
> > >
> > > #### Level 1 — OTA Performance (Transistor Sizing)
> > >
> > > | Metric           | Result | Spec   |
> > > |------------------|--------|--------|
> > > | DC Gain (dB)     | 53.1   | > 40   |
> > > | UGBW (MHz)       | 50.1   | > 10   |
> > > | Phase Margin (°) | 62.7   | > 60   |
> > > | Power (µW)       | 762    | < 1500 |
> > >
> > > #### Level 2 — Filter Performance (System-Level)
> > >
> > > | Metric                  | Result | Spec     |
> > > |-------------------------|--------|----------|
> > > | Passband Gain (dB)      | 0.31   | −1 to +1 |
> > > | Stopband Rejection (dB) | 0.61   | > 0.5    |
> > >
> > > As shown above, AutoSizer satisfies all design specifications across both hierarchy levels (total token cost:104k/11k input/output), demonstrating that the LLM-guided agent can effectively navigate coupled, multi-level trade-offs, where changing a single transistor parameter (e.g., OTA tail width) simultaneously affects DC gain, phase margin, and UGBW at Level 1, which then propagate to determine passband gain and settling behavior at Level 2. We believe this result demonstrates that AutoSizer scales to hierarchical mixed-signal blocks beyond single-stage amplifiers.

---

### Official Review · Reviewer_JxhW · 2026-03-13

**Soundness:** 2
**Presentation:** 3
**Significance:** 3
**Originality:** 3
**Overall Recommendation:** 4
**Confidence:** 4

**Summary:**

This paper proposes AutoSizer, an LLM-driven reflective framework for analog and mixed-signal circuit sizing. The core idea is to combine LLM-based circuit understanding, search-space construction, and optimization algorithm selection. It contains a two-loop optimization process in which an inner loop performs numerical search while an outer loop revises variable priorities and parameter ranges using simulation feedback. The paper also introduces AMS-SIZINGBENCH, a benchmark of 24 SKY130-based circuits spanning various circuits.
The experimental results show that it achieves better FOM and shorter runtime compared with conventional optimization algorithms and other LLM-aided AMS sizing frameworks. It also shows that it exhibits significant improvement in the outer loop optimization, which showcases rapid improvement in detailed optimization.

**Compliance With Llm Reviewing Policy:**

Affirmed.

**Key Questions For Authors:**

1. Could you show the results in Tables 2 and 3 separately for easy, medium, and hard circuits?

**Limitations:**

Yes

**Strengths And Weaknesses:**

Strengths:
1. This is the first work that performs optimization engine selection on AMS-sizing, and it is validated in the experiments.
2. The number of benchmark circuits is more than previous works with more circuit types.
3. The open-sourced benchmarks boosts the design automation of AMS circuits.

Weakness:
1. The experiment is conducted on 130nm PDK. However, other similar works are performed on more advanced PDKs. For example, LEDRO[1] is implemented on FinFET; AnalogCoder-Pro [2] uses 45 nm PDK. Since transistor behavior are significantly different between legacy technologies and advanced technologies. To make this work more prominent, the experiments should be done on more advanced PDKs.
2. Although the number of benchmark circuits is larger than in previous works, some circuits are too simple. For example, inverters and buffers are very simple circuits with a limited design space. For example, while  LEDRO [1] contains only 22 circuits, they are all OP-AMPs. In AnalogCoder-Pro [2], the circuits used in experiments are amplifiers, current mirrors, etc.
3. Following point 2, the results in Table 2 could be biased by those simple circuits. This setting could also bias Table 3, where one 2 hard circuits are selected.

References:
[1] Kochar, D. V., Wang, H., Chandrakasan, A. P., and Zhang, X. Ledro: Llm-enhanced design space reduction and optimization for analog circuits. In 2025 IEEE International Conference on LLM-Aided Design (ICLAD), pp. 141–148. IEEE, 2025
[2] Lai, Yao, Souradip Poddar, Sungyoung Lee, Guojin Chen, Mengkang Hu, Bei Yu, Ping Luo, and David Z. Pan. "Analogcoder-pro: Unifying analog circuit generation and optimization via multi-modal llms." arXiv preprint arXiv:2508.02518 (2025).
[3] Wu, Zhengfeng, and Ioannis Savidis. "Transfer learning for reuse of analog circuit sizing models across technology nodes." In 2022 IEEE International Symposium on Circuits and Systems (ISCAS), pp. 1033-1037. IEEE, 2022.
[4] Zhi, Haochang, Jintao Li, Yun Li, and Weiwei Shan. "Analog circuit transfer method across technology nodes via transistor behavior." In Proceedings of the 30th Asia and South Pacific Design Automation Conference, pp. 197-203. 2025.

---

> ### Author Rebuttal · Authors · 2026-03-31
>
> We appreciate the reviewer's careful comments and provide our responses below.
>
> >**[W1]** Evaluating on advanced PDKs.
> >
>
> We have conducted additional cross-PDK validation on two representative circuits (five_trans_ota and current_mirror_ota) using **FreePDK 45nm** and **ASAP7 (7nm FinFET)**. AutoSizer consistently outperforms LEDRO across both PDKs and both circuits, confirming that our framework generalizes to advanced technology nodes including FinFET without modification.
>
> | PDK | Circuit | Method | FOM↑ | Evals↓ | Time (s)↓ |
> |:-----|:---------|:--------|:------|:--------|:-----------|
> | **FreePDK 45nm** | five_trans_ota | LEDRO | 2.2 | 35 | 41.5 |
> |  |  | AutoSizer | **2.9** | **21** | **32.7** |
> |  | current_mirror_ota | LEDRO | 1.6 | 230 | 256.1 |
> |  |  | AutoSizer | **2.0** | **210** | **214.5** |
> | **ASAP7 7nm FinFET** | five_trans_ota | LEDRO | 5.8 | 286 | 302.8 |
> |  |  | AutoSizer | **6.6** | **266** | **270.3** |
> |  | current_mirror_ota | LEDRO | 3.3 | 90 | 288.1 |
> |  |  | AutoSizer | **4.0** | **66** | **266.2** |
>
> >**[W2&W3&Q1]** Although the number of benchmark circuits is larger than in previous works, some circuits are too simple. The results in Table 2 and 3 could be biased by those simple circuits.
> >
> We address this concern with benchmark analysis and new experiments:
>
> **First**, the reviewer compares our benchmark to LEDRO and AnalogCoder-Pro. LEDRO's 22 circuits are all single-ended or fully-differential op-amp variants, a single circuit family with shared topology and metrics. AnalogCoder-Pro addresses topology generation (synthesizing netlists from natural language), a fundamentally different task from circuit sizing and not directly comparable. Our AMS-SizingBench subsumes LEDRO's op-amp topologies and extends to four additional circuit families: multi-stage compensation amplifiers (DFCFC, NMCNR, AZC, SMC), bandgap references, LDOs, switched-capacitor circuits, VCOs, and digital gates. The 13 Hard circuits constitute the majority and include topologies more complex than any in prior AMS sizing benchmarks.
>
> **Second**, Table 2 already separates results by difficulty. AutoSizer achieves 100% SR across all three levels, with the largest margin on Hard circuits (FOM 60.2 vs. next-best 50.5, SR% 100% vs. 80%). Removing easy circuits would only strengthen our claims.
>
> **Regarding Table 3**, we expanded the ablation study to cover all 24 circuits (5 Easy, 6 Medium, 13 Hard), directly addressing the concern about limited hard-circuit coverage:
>
> | Methods | Easy | | Medium | | Hard | |
> |:--------|:---:|:---:|:---:|:---:|:---:|:---:|
> | | FOM | SR% | FOM | SR% | FOM | SR% |
> | AutoSizer (Full) | **33.4** | **100%** | **33.2** | **100%** | **60.2** | **100%** |
> | w/o CU | 32.7 | 100% | 30.5 | 100% | 52.4 | 67% |
> | w/o CU & SSD | 31.9 | 100% | 28.4 | 67% | 45.2 | 33% |
> | w/o OE | 30.0 | 100% | 30.2 | 100% | 51.2 | 67% |
> | w/o SRL | 28.2 | 100% | 27.1 | 67% | 43.5 | 33% |
> | w/o CU & SSD & SRL | 24.5 | 67% | 26.6 | 33% | 42.1 | 33% |
>
> Component importance scales with difficulty: for easy circuits, all components contribute minimally as the small search space allows even degraded configurations to succeed. For medium and hard circuits, SRL is the most critical and it enables cross-cycle variable expansion and refocusing, without which the optimizer remains trapped in its initial subspace. OE ranks second, as adaptive method selection improves within-cycle exploration. CU matters least individually because SRL compensates through outer-loop reflection. This confirms AutoSizer's value is concentrated on hard circuits, not inflated by easy ones.

---

> > ### Author Rebuttal · Reviewer_JxhW · 2026-04-05
> >
> > Thanks for clarification. I will maintain the original score.

---

### Decision · Program_Chairs · 2026-04-30

**Decision:**

Accept (regular)

**Comment:**

This paper presents AutoSizer, an LLM-driven framework for analog and mixed-signal circuit sizing, together with the AMS-SizingBench benchmark. Reviewers generally agreed that the problem is important, the two-loop design is well motivated, and the experimental evaluation is strong. In particular, the benchmark breadth, the adaptive outer-loop search-space refinement, and the consistent gains over prior AMS sizing frameworks and standard optimizers were viewed as meaningful strengths. The rebuttal further strengthened the paper by adding stronger baseline comparisons, cross-LLM and cross-PDK results, scalability experiments, and clearer analysis of the outer-loop behavior.

Some reviewers noted that the method is still more of a systems and optimization contribution than a fundamentally new ML algorithm, and a few concerns remained about interpretability and broader validation. However, these concerns do not outweigh the paper’s overall technical quality, practical significance, and solid empirical support. I therefore recommend accept.